# Discovering Latent Structural Causal Models from Spatio-Temporal Data

## Abstract

Many important phenomenon in scientific fields such as climate, neuroscience and epidemiology are naturally represented as spatiotemporal gridded data with complex interactions. Inferring causal relationships from these data is a difficult problem compounded by the high dimensionality of such data and the correlations between spatially proximate points. We present SPACY (SPAtiotemporal Causal discoverY), a novel framework based on variational inference, designed to explicitly model latent time-series and their causal relationships from spatially confined modes in the data. Our method uses an end-to-end training process that maximizes an evidence-lower bound (ELBO) for the data likelihood. Theoretically, we show that, under some conditions, the latent variables are identifiable up to transformation by an invertible matrix. Empirically, we show that SPACY outperforms state-of-the-art baselines on synthetic data, remains scalable for large grids, and identifies key known phenomena from real-world climate data.

## 1 Introduction

In several scientific domains such as climate science, neurology, and epidemiology, low-level sensor measurements generate high-dimensional observational data. These data are naturally represented as gridded time series, with interactions that evolve over both space and time. Discovering causal relationships from spatiotemporal gridded time-series data is an important scientific task that allows researchers to predict future states, intervene in harmful trends, and develop new insights into

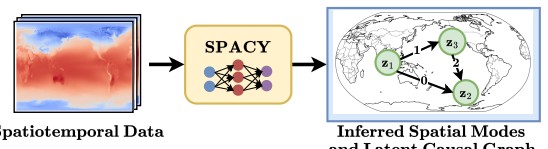

**Spatiotemporal Data**  **Inferred Spatial Modes and Latent Causal Graph**

Figure 1: SPACY jointly infers latent time series and the underlying causal graph from gridded time-series data by identifying spatial modes of variability.

the underlying mechanisms. In climate science, the study of teleconnections (Liu et al., 2023), the interactions between regions thousands of kilometers away, is important to understanding how climate events in one part of the world may affect weather patterns in distant locations.

Several methods have been developed for causal structure learning from time-series data (Granger, 1969; Hyvärinen et al., 2010; Runge, 2020a; Tank et al., 2021; Gong Wenbo & Nick, 2022; Cheng et al., 2023). However, applying these methods to spatiotemporal data presents significant challenges. The high dimensionality of large gridded data makes it difficult for many of these techniques, especially those relying on conditional independence tests, to scale effectively (Glymour et al., 2019). Additionally, spatially proximate points often exhibit highly correlated, redundant time series. Conditioning on nearby correlated points can obscure true causal relationships between distant locations, reducing statistical power and leading to inaccurate results (Tibau, 2022).

Recent advances in spatiotemporal causal discovery have sought to address these challenges. One common approach is a two-stage process: first, dimensionality reduction is applied to extract a small number of latent time series from the original grid of time series; then, causal discovery is performed on these reduced-dimensional representations. Examples of this approach include Tibau (2022) and Falasca et al. (2024). However, these methods perform dimensionality reduction independent of the causal structure, potentially leading to low-dimensional representations that obscure

the relationships among causally relevant entities. Another important line of research is causal representation learning from time series data (Schölkopf et al., 2021). While approaches like those in Yao et al. (2022b;a); Chen et al. (2024) model latent time series from high-dimensional data, they do not incorporate spatial priors, making them less suitable for spatiotemporal causal discovery. Causal Discovery with Single-parent Decoding (CDSD) (Brouillard et al., 2024; Boussard et al., 2023) learns a mapping from the observational time series to latent variables to infer the latent time series. However, it assumes that each observed variable is influenced by only one latent variable.

We present a novel variational inference-based framework for spatial-temporal causal discovery called SPAtio-temporal Causal DiscoverY (SPACY) to address these limitations (Figure 1). Our approach jointly infers both the latent time series and the underlying causal graph in an end-to-end process. The key idea of our approach is to learn the location and scale parameters of spatial factors on the grid, which we model using Radial Basis Functions (RBFs). These spatial factors determine the grid locations corresponding to each inferred latent time series. Additionally, we analyze the identifiability of our framework. We demonstrate that when the grid is infinitely fine, we can uniquely recover the spatial factors and latent time series (up to permutation) that generate the observed data distribution. Notably, compared to previous works, our framework can handle both instantaneous edges and overlapping spatial factors, allowing observed variables to be associated with multiple latent factors.

Our main contributions can be summarized as follows:

1. We introduce SPAtio-temporal Causal discoverY (SPACY), a novel variational inference-based causal discovery framework that tackles realistic and challenging settings of spatiotemporal datasets by simultaneously inferring the latent causal representation time series and the underlying causal graph.

2. Theoretically, we show that, under some conditions, the latent factors are identifiable up to transformation by an invertible matrix from the observational data when the resolution of the grid is infinite.

3. Experimentally, we demonstrate the strong performance of our method on both synthetic and real-world datasets. SPACY can infer both lagged and instantaneous causal links from high-dimensional grids in a tractable manner.

## 2 RELATED WORK

In this section, we provide an overview of the literature on causal discovery from time-series, causal representation learning and spatiotemporal causal discovery.

**Causal Discovery from time series data.** A prominent line of research in time series causal discovery is based on Granger causality, as introduced by Granger (1969). For example, Tank et al. (2021) use component-wise MLPs and LSTMs with sparsity constraints to infer non-linear Granger causality. In contrast, Khanna & Tan (2020) apply Statistical Recurrent Units (SRUs) to detect causal relationships across multiple scales. Löwe et al. (2022) propose Amortized Causal Discovery using a variational autoencoder and Graph Neural Networks. Cheng et al. (2023; 2024) infers Granger causal links from irregularly sampled or incomplete data by simultaneously imputing missing values. However, Granger causality only captures predictive relationships and ignores instantaneous effects, latent confounders, and history-dependent noise (Peters et al., 2017).

The Structural Causal Model (SCM) framework can theoretically overcome these limitations by explicitly modeling the causal relationships between variables. Hyvärinen et al. (2010) extend LiNGAM (Shimizu et al., 2006) to develop VARLiNGAM, incorporating vector autoregressive models for time series data. DYNOTEARS (Pamfil et al., 2020) adapts NOTEARS (Zheng et al., 2018) to dynamic Bayesian networks. Both methods, however, are restricted to linear relationships. PCMCI and PCMCI$^+$ (Runge et al., 2019; Runge, 2020a) extend the PC (Spirtes et al., 2000) algorithm to handle instantaneous effects. Rhino (Gong Wenbo & Nick, 2022) uses neural networks to model the functional relationships and estimates the temporal adjacency matrix from observational data while accounting for exogenous history-dependent noise distributions. Wang et al. (2024) use stochastic differential equations for causal structure learning from continuous-time temporal processes with potentially irregular sampling.

However, applying these methods directly to spatiotemporal data presents significant challenges. The high dimensionality of large gridded datasets makes it difficult for many techniques—especially those that rely on conditional independence tests—to scale effectively (Glymour et al., 2019). Furthermore, spatially proximate points often exhibit highly correlated and redundant time series. Conditioning on these nearby correlated points can obscure true causal relationships between distant locations, reducing statistical power and leading to inaccurate results (Tibau, 2022).

**Spatiotemporal Causal Discovery/Causal Representation Learning** Numerous studies have extended Granger causality to spatiotemporal settings, particularly in climate science (Mosedale et al., 2006; Kodra et al., 2011; Ali et al., 2024). Lozano et al. (2009) proposed a method combining Granger causality with a Group Elastic Net to capture spatial and temporal dependencies, enabling the identification of causal relationships among climate variables. However, the model assumes that the causal relationships are linear, and only infers a summary graph.

One approach to spatiotemporal causal discovery is to perform dimensionality reduction to obtain a smaller number of latent time series and then infer a causal graph among the latent variables. For example, Tibau (2022) use Varimax for dimensionality reduction and PCMCI$^+$ (Runge, 2018; 2020b) for causal discovery. Falasca et al. (2024) infer regional modes based on correlation and spatial proximity, applying linear-response theory to uncover causal links. A key limitation of these methods is that dimensionality reduction occurs independently of the causal structure in the data. Consequently, the latent variables may not correspond to causally relevant entities. Additionally, conditional independence-based methods are computationally intensive as they may require an exponential number of conditional independence tests.

Other approaches use neural networks to model nonlinear interactions. The Spatial-Temporal Causal Discovery Framework (STCD) (Sheth et al., 2022) utilizes attention-based convolutional neural networks to identify causal relationships from gridded time-series data. However, it encodes an explicit form of spatial dependence specific to the problem of hydrological systems (i.e. reduce attention scores based on geographic height) rather than inferring it from data.

Causal representation learning from time series involves inferring abstract, high-level causal variables and their relationships from temporal data. Lippe et al. (2022; 2023) focus on causal representation learning from interventional time-series data. Yao et al. (2022b;a); Chen et al. (2024) introduce frameworks to recover latent causal variables and identify their relations from observational sequential data. However, these methods do not model instantaneous edges in the causal graph. Morioka & Hyvarinen (2024) prove the identifiability of causal relations, even in the presence of instantaneous edges, by assuming that the observational variables can be appropriately grouped. However, this grouping is rarely known in practice apriori. Moreover, none of these methods consider the spatial structure present in the data. The work most closely related to ours is Causal Discovery with Single Parent Decoding (CDSD) (Brouillard et al., 2024; Boussard et al., 2023), which learns a mapping from the observational time series to latent variables. However, CDSD operates under the assumption that each observed variable is influenced by only one latent variable, and the causal graph has no instantaneous edges. In contrast, SPACY allows both instantaneous edges and overlapping modes, i.e., an observational variable can be influenced by more than one latent variable.

**Preliminaries.** A Structural Causal Model (Pearl, 2009) (SCM) explicitly defines the causal relationships between variables in the form of functional equations. Formally, an SCM over $D$ variables consists of a 5-tuple $\langle \mathcal{X}, \varepsilon, \mathcal{F}, \mathcal{G}, P(\varepsilon) \rangle$:

1. Endogenous (observed) variables $\mathcal{X} = \{X^1, X^2, \dots, X^D\}$;

2. Exogenous (noise) variables $\varepsilon = \{\varepsilon^1, \varepsilon^2, \dots, \varepsilon^D\}$ influencing the endogenous variables.

3. A *Directed Acyclic Graph* (DAG) $\mathbf{G}$, denoting the causal links amongst the members of $\mathcal{X}$;

4. A set of $D$ functions $\mathcal{F} = \{f^1, f^2, \dots, f^D\}$ determining $\mathcal{X}$ through the equations $X^i = f^i(\text{Pa}_{\mathbf{G}}^i, \varepsilon^i)$, where $\text{Pa}_{\mathbf{G}}^i \subset \mathcal{X}$ denotes the parents of node $i$ in graph $\mathbf{G}$ and $\varepsilon^i \subset \varepsilon$;

5. $P(\varepsilon)$, which describes a distribution over noise $\varepsilon$.

## 3 SPACY: SPATIAL-TEMPORAL CAUSAL DISCOVERY

**Problem Setting.** We are given $N$ samples of $L$-dimensional multivariate time series with $T$ timesteps each. These $L$ time series are arranged in a $K$-dimensional grid $\mathcal{G}$. In our setting, we consider $K = 2$, i.e. a two-dimensional grid. We denote the observational time series as $\left\{ \mathbf{X}_{1:L}^{(1:T),n} \right\}_{n=1}^{N}$. We assume that the dynamics of the observed data are driven by interactions in a smaller number of *latent* (i.e. unobservable) time series. We denote the $D$ latent time series for each of the $N$ samples as $\left\{ \mathbf{Z}_{1:D}^{(1:T),n} \right\}_{n=1}^{N}$, with $D << L$. The latent time series is stationary with a maximum time lag of $\tau$, meaning the present is influenced by up to $\tau$ past timesteps. Interactions in the latent time series follow an SCM represented by a DAG $\mathbf{G}$. Our goal is to infer the latent time series $\left\{ \mathbf{Z}_{1:D}^{(1:T),n} \right\}_{n=1}^{N}$ and the causal graph $\mathbf{G}$ in an unsupervised manner.

### 3.1 FORWARD MODEL

We formalize our assumptions about the data generation process using a probabilistic graphical model (Figure 2). We assume that the latent time series $\mathbf{Z}$ is generated by an SCM with causal graph $\mathbf{G}$. The number of latent variables $D$ is input as a hyperparameter. The spatial correlations between nearby grid points are captured by the spatial factors $\mathbf{F} \in \mathbb{R}^{L \times D}$, parameterized by $\boldsymbol{\rho}$ and $\boldsymbol{\gamma}$. These factors map the latent time series $\mathbf{Z}_{1:D}^{(1:T)} \in \mathbb{R}^{D \times T}$ to the observed time series $\mathbf{X}_{1:L}^{(1:T)} \in \mathbb{R}^{L \times T}$.

**Latent SCM.** We model the latent SCM that describes the dynamics of $\mathbf{Z}^{(t)}$ as an additive noise model (Hoyer et al., 2008):

$$\mathbf{Z}_d^{(t)} = f_d \left( \text{Pa}_{\mathbf{G}}^d(< t), \text{Pa}_{\mathbf{G}}^d(t) \right) + \eta_d^{(t)}$$

The causal graph $\mathbf{G}$ specifies the causal parents of each node, represented by a temporal adjacency matrix with shape $(L+1) \times D \times D$. The parent nodes from previous and current time steps are denoted by $\text{Pa}_{\mathbf{G}}^d(< t)$ and $\text{Pa}_{\mathbf{G}}^d(t)$ respectively. We assume that $\mathbf{Z}_d^t$ is influenced by at most $\tau$ preceding time steps, i.e., $\text{Pa}_{\mathbf{G}}(< t) \subseteq \{\mathbf{Z}^{t-1}, \ldots, \mathbf{Z}^{t-\tau}\}$. $\mathbf{G}^{1:\tau}$ represents the lagged relationships and $\mathbf{G}^0$ represents the instantaneous edges. The time-lag $\tau$ is treated as a hyperparameter.

We implement two variants of SPACY based on the type of functional relationships being modeled.

**SPACY-L.** This variant models linear relationships with independent noise. $f_d$ is defined as:

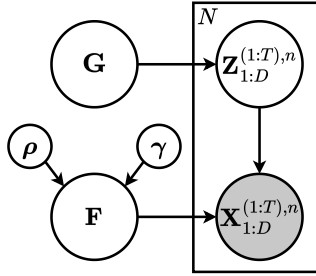

Figure 2: Probabilistic graphical model for SPACY. Shaded circles are observed and hollow circles are latent.

$$f_d \left( \text{Pa}_{\mathbf{G}}^d(\le t) \right) = \sum_{k=0}^{\tau} \sum_{d'=1}^{D} (\mathbf{G} \circ W)_{d',d}^{k} \times \mathbf{Z}_{d'}^{t-k}, \tag{1}$$

where $\circ$ denotes the Hadamard product, and $W \in \mathbb{R}^{(\tau+1) \times D \times D}$ is a learned weight tensor. We assume that $\eta_d^t$ is isotropic Gaussian noise.

**SPACY-NL.** This variant models non-linear relationships using Rhino (Gong Wenbo & Nick, 2022), which accounts for both instantaneous effects and history-dependent noise. We parameterize the structural equations $f_d$ using MLPs $\xi_f$ and $\lambda_f$ shared across all nodes. We use trainable embeddings $\mathcal{E} \in \mathbb{R}^{(\tau+1) \times D \times D}$ with embedding dimension $e$ to distinguish between nodes. $f_d$ is defined as:

$$f_d \left( \text{Pa}_{\mathbf{G}}(\le t) \right) = \xi_f \left( \sum_{k=0}^{\tau} \sum_{j=1}^{D} \mathbf{G}_{j,d}^{k} \times \lambda_f \left( \left[ \mathbf{Z}_j^{t-k}, \mathcal{E}_j^k \right], \mathcal{E}_0^d \right) \right). \tag{2}$$

The noise model is based on conditional spline flows (Durkan et al., 2019), with the parameters of the spline flow predicted by MLPs $\xi_\eta$ and $\lambda_\eta$, which share a similar architecture to $\xi_f$ and $\lambda_f$.

**Spatial Factors.** The low-dimensional latent time series are mapped to the high-dimensional grid by the spatial factors $\mathbf{F} \in \mathbb{R}^{L \times D}$. The $d^{\text{th}}$ column of $\mathbf{F}$ represents the influence of the $d^{\text{th}}$ latent

Figure 3: Overview of the ELBO calculation for SPACY. The model processes spatiotemporal data $\left\{\mathbf{X}_{1:L}^{(1:T),n}\right\}_{n=1}^{N}$ to infer latent time series $\left\{\mathbf{Z}_{1:D}^{(1:T),n}\right\}_{n=1}^{N}$, where $D \ll L$. Causal relationships are modeled using a DAG $\mathbf{G}$ sampled from $q_\phi(\mathbf{G})$. Latent time-series are mapped to grid locations via spatial factors $\mathbf{F}$ sampled from $q_\phi(\mathbf{F})$. Arrows in $\mathbf{G}$ are labeled with edge time-lags.

variable on each grid location. To effectively capture the correlation between spatially proximate grid points under a single latent variable, we model the spatial factors using radial basis functions (RBFs), following Manning et al. (2014); Farnoosh & Ostadabbas (2021). RBFs not only ensure locality, they are also smooth functions that are parameter-efficient. We assume a uniform prior over the grid $\mathcal{G}$ for the center parameter $\boldsymbol{\rho}_d$ of each kernel, and assume that the scale parameter $\boldsymbol{\gamma}_d$ comes from a standard normal distribution. Mathematically,

$$\boldsymbol{\rho}_d \sim U[0,1]^K, \boldsymbol{\gamma}_d \sim \mathcal{N}(0, I), \tag{3}$$

$$\mathbf{F}_d^\ell = \text{RBF}_d(x_\ell; \boldsymbol{\rho}_d, \boldsymbol{\gamma}_d) = \exp\left(-\frac{||x_\ell - \boldsymbol{\rho}_d||^2}{\exp(\boldsymbol{\gamma}_d)}\right), \tag{4}$$

where $x_\ell$ refers to the spatial coordinates of the $\ell^{\text{th}}$ grid point.

The observational time series is assumed to be generated by applying a grid point-wise non-linearity $g_\ell$ to the product of the spatial factors and latent time series, with additive Gaussian noise. We implement the nonlinearity $g_\ell$ as an MLP $\Xi$ shared across all grid-points, with concatenated embeddings $\mathscr{G} \in \mathbb{R}^{L \times f}$, where $f$ is the embedding dimension. In equations,

$$\mathbf{X}_\ell^{(t)} = g_\ell\left([\mathbf{FZ}]_\ell^{(t)}\right) + \varepsilon_\ell^{(t)}, \quad \varepsilon_\ell^{(t)} \sim \mathcal{N}(0, \sigma_\ell^2 I) \tag{5}$$

$$g_\ell(x) = \Xi\left([x, \mathscr{G}_\ell]\right), \quad \mathscr{G}_\ell \in \mathbb{R}^f \tag{6}$$

## 3.2 VARIATIONAL INFERENCE

Let $\theta$ denote the parameters of the forward model. Ideally, we would estimate $\theta$ using maximum likelihood estimation. However, the likelihood $p_\theta(\mathbf{X})$ is intractable due to the presence of latent variables $\mathbf{Z}$, $\mathbf{G}$ and $\mathbf{F}$. To address this, we propose using variational inference, optimizing an evidence lower bound (ELBO) instead.

**Proposition 1.** *The data generation model described in Figure 2 admits the following evidence lower bound (ELBO):*

$$
\log p_\theta \left( \mathbf{X}^{(1:T),1:N} \right) \geq \sum_{n=1}^{N} \left\{ \mathbb{E}_{q_\phi(\mathbf{Z}^{(1:T),n}|\mathbf{X}^{(1:T),n})q_\phi(\mathbf{G})q_\phi(\mathbf{F})} \left[ \log p_\theta \left( \mathbf{X}^{(1:T),n}|\mathbf{Z}^{(1:T),n}, \mathbf{F} \right) \right. \right.
$$

$$
\left. \left. + \left[ \log p_\theta \left( \mathbf{Z}^{(1:T),n}|\mathbf{G} \right) - \log q_\phi(\mathbf{Z}^{(1:T),n}|\mathbf{X}^{(1:T),n}) \right] \right] \right\} + \mathbb{E}_{q_\phi(\mathbf{G})}[\log p(\mathbf{G}) - \log q_\phi(\mathbf{G})]
$$

$$
+ \mathbb{E}_{q_\phi(\mathbf{F})}[\log p(\mathbf{F}) - \log q_\phi(\mathbf{F})] = ELBO(\theta, \phi) \tag{7}
$$

See section A.1.1 for the derivation. We outline the computation of the ELBO in Figure 3. $q_\phi$ represents the variational distribution, with variational parameters $\phi$. The first term $\log p_\theta(\mathbf{X}^{(1:T),n}|\mathbf{Z}^{(1:T),n}, \mathbf{F})$ in equation 7 represents the conditional likelihood of the observed data $\mathbf{X}^{(1:T),n}$ conditioned on $\mathbf{Z}^{(1:T),n}$ and $\mathbf{F}$, and represents how well the observed data is fit. The remaining terms represent the KL divergences of the variational distributions from their prior distributions. More details about the implementation of the loss terms are in Appendix A.2.

We detail the implementation of the variational distributions below:

**Causal graph** $q_\phi(\mathbf{G})$. The variational distribution for the adjacency matrix $q_\phi(\mathbf{G})$ is modeled as a product of independent Bernoulli distributions, indicating the presence or absence of every edge. To compute the expectation over $q_\phi(\mathbf{G})$, we sample one graph using Monte Carlo sampling, leveraging the Gumbel-Softmax trick (Jang et al., 2017).

**Spatial Factor** $q_\phi(\mathbf{F})$. We model the variational distributions of the center and scale parameters $\boldsymbol{\rho}_d$ and $\boldsymbol{\gamma}_d$ as normal distributions with learnable mean and log-variance parameters $(\boldsymbol{\mu}_{\boldsymbol{\rho}_d}, v_{\boldsymbol{\rho}_d}), (\boldsymbol{\mu}_{\boldsymbol{\gamma}_d}, v_{\boldsymbol{\gamma}_d})$. To sample from $q_\phi(\mathbf{F})$, we first sample $\boldsymbol{\rho}_d$ and $\boldsymbol{\gamma}_d$ using the reparameterization trick (Kingma & Welling, 2014), and then compute the RBF kernel using these parameters. To ensure that the coordinates of the center lie in the range $[0, 1]$, we apply the sigmoid function.

$$
\boldsymbol{\rho}_d \sim \mathcal{N} \left( \boldsymbol{\mu}_{\boldsymbol{\rho}_d}, \exp \left( v_{\boldsymbol{\rho}_d} \right) I \right), \boldsymbol{\gamma}_d \sim \mathcal{N} \left( \boldsymbol{\mu}_{\boldsymbol{\gamma}_d}, \exp \left( v_{\boldsymbol{\gamma}_d} \right) I \right)
$$

$$
\mathbf{F}_d^\ell = \mathrm{RBF}_d(x_\ell; \boldsymbol{\rho}_d, \boldsymbol{\gamma}_d) = \exp \left( -\frac{||x_\ell - \mathrm{sigmoid}(\boldsymbol{\rho}_d)||^2}{\exp(\boldsymbol{\gamma}_d)} \right).
$$

**Encoder** $q_\phi(\mathbf{Z}^{(1:T),n}|\mathbf{X}^{(1:T),n})$. To obtain the latents from the observational samples, we use a neural network encoder. Specifically, the variational distribution $q_\phi(\mathbf{Z}^{(1:T),n}|\mathbf{X}^{(1:T),n})$ is modeled as a normal distribution whose mean and log-variance are output by MLPs $\zeta_\mu$ and $\zeta_{\sigma^2}$. We sample $\mathbf{Z}$ from the distribution using the reparameterization trick:

$$
\mathbf{Z}^{(t),n} \sim \mathcal{N} \left( \zeta_\mu(\mathbf{X}^{(t),n}), \exp \left( \zeta_{\sigma^2}(\mathbf{X}^{(t),n}) \right) \right).
$$

## 4 IDENTIFIABILITY ANALYSIS

In this section, we examine the identifiability of the generative model introduced in Section 3.1. Roughly speaking, a model is said to be identifiable if the latent variables can be uniquely recovered from observational data. Several prior works have investigated the identifiability of latent parameters in various deep generative models (Khemakhem et al., 2020; Zheng et al., 2022; Yao et al., 2022b).

We focus on the specific case where no non-linearity maps the latents to the observable space, meaning $g_\ell$ in equation 6 is the identity map. To analyze identifiability, we extend the notion of a gridded time series to infinite resolution. Instead of observing the time series at a finite set of grid points, we assume it can be observed at every point within the bounded $K$-dimensional grid $\mathcal{G} = [0, 1]^K$. In this framework, $\mathbf{X}(x)$ represents a $T$-dimensional random variable describing the observational time series at location $x$ on the grid.

We also generalize our assumptions about how the spatial factors are generated, and assume that they are function evaluations at the grid points of a family of linearly independent functions. Notably, the family of RBF functions are one such family of functions (Smola & Schölkopf, 1998). To formalize this, we introduce the following definition.

**Definition** (Spatial Factor Process). Let $\mathcal{G} = [0, 1]^K$ be a $K-$dimensional grid, and let $\mathbf{Z} \in \mathbb{R}^{D \times T}$. Suppose $\mathcal{F} = \{F_{\psi_1}, ..., F_{\psi_D}\}$ is a finite linearly independent family. We define a Spatial Factor Process SFP($\mathbf{Z}, \mathcal{F}, p_\varepsilon$), denoted by $\mathbf{X} : \mathcal{G} \to \mathbb{R}^T$, as follows: for each location $x \in \mathcal{G}$ in the grid,

$$\mathbf{X}(x) = \mathbf{F}_x^\top \mathbf{Z} + \varepsilon_x, \text{ where } \mathbf{F}_x = [F_{\psi_1}(x), \quad \ldots, \quad F_{\psi_D}(x)]^\top \tag{8}$$

and $\varepsilon_x \sim p_\varepsilon(\cdot)$ is a normally distributed noise term.

The first result demonstrates that if two SFPs are equal at all grid points, they must share the same spatial factors and latent time series, up to a permutation. This implies that the spatial factors and latent time series are identifiable from the *conditional* likelihood $\log p_\theta(\mathbf{X}(x)|\mathbf{Z}, \mathbf{F}_x)$.

**Theorem 1** (Identifiability of SFPs). *Given two SFPs* $\mathbf{X} = SFP(\mathbf{Z}, \mathcal{F}, p_\varepsilon)$ *and* $\mathbf{Y} = SFP(\tilde{\mathbf{Z}}, \tilde{\mathcal{F}}, q_\varepsilon)$ *where none of the rows of* $\mathbf{Z}$ *or* $\tilde{\mathbf{Z}}$ *are all zero, such that* $p(\mathbf{X}(x)) = p(\mathbf{Y}(x))$ *for every* $x \in \mathcal{G}$, *then* $\mathbf{Z} = P\tilde{\mathbf{Z}}$ *and* $\mathcal{F} = \tilde{\mathcal{F}}$ *for some permutation matrix* $P$.

We now turn to the identifiability of the latent time series from the observational distribution. The following result shows that the latent variable distribution can be recovered up to a transformation by an invertible matrix. Although not as precise as Theorem 1, it still guarantees that the latents are partially identifiable.

**Theorem 2** (Identifiability of the latents). *Suppose two spatial factor processes* $\mathbf{X}(x)$ *and* $\tilde{\mathbf{X}}(x)$ *with spatial factors* $\mathbf{F}_x$ *and* $\tilde{\mathbf{F}}_x$ *have the same observational distributions for all* $x \in \mathcal{G}$. *Then the latent variable distribution is identifiable up to transformation by an invertible matrix.*

The detailed mathematical statements and proofs for these results are provided in Appendix A.1.2.

## 5 EXPERIMENTS

We assess SPACY's ability to capture causal relationships across various spatiotemporal contexts using both synthetic datasets with known ground truth and simulated climate datasets. Our results demonstrate that SPACY consistently uncovers accurate causal relationships while generating interpretable outputs. An implementation of SPACY is available at (https://anonymous.4open.science/r/spacy-572B/). The code is built with PyTorch 2.1 and run on machines with NVIDIA A10 GPUs.

**Baselines.** We compare SPACY with state-of-the-art baselines. We include the two-step algorithms Mapped PCMCI (Varimax-PCA + PCMCI$^+$ with Partial Correlation test) (Tibau, 2022; Runge, 2020b) and the Linear Response method (Falasca et al., 2024). We also evaluate against the causal representation learning approaches, LEAP (Yao et al., 2022b) and TDRL (Yao et al., 2022a).

### 5.1 SYNTHETIC DATA

**Setup.** Since real-world datasets lack ground truth causal graphs, we generate synthetic datasets with known causal relationships to benchmark SPACY's causal discovery performance. These are generated from randomly constructed ground-truth graphs, following the forward model described in Figure 2. We experiment with several configurations of synthetic data. The latent time series are generated using either (1) a linear structural causal model (SCM) with randomly initialized weights and additive Gaussian noise, or (2) a nonlinear SCM, where the structural equations are modeled by randomly initialized MLPs, combined with additive history-dependent conditional-spline noise.

The mapping function $g_\ell$ is set as (1) linear, where the identity function is used, or (2) nonlinear, where an MLP is used. For each configuration, we generate $N = 100$ samples, each with time length $T = 100$ and a grid of size $100 \times 100$ ($L = 10^4$). Datasets are generated with $D = 10, 20$ and 30 nodes in each setting. For more details on dataset generation, refer to Appendix A.3.1.

We assess the performance of SPACY and the baselines using two metrics: the orientation F1 score of the inferred causal graph $\mathbf{G}$, and the mean correlation coefficient (MCC) between the learned and ground-truth latent representations $\mathbf{Z}$. More details on the evaluation process are in Appendix A.2.4.

**Results.** The results of the synthetic experiments are shown in Figure 4. SPACY consistently outperforms all other methods across all settings of $D$ in terms of F1 score. On the linear SCM

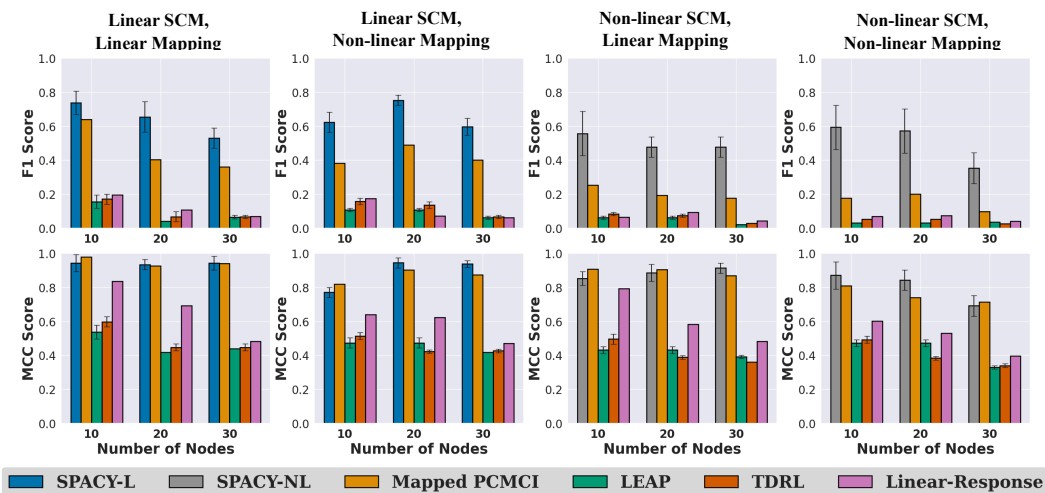

Figure 4: Results on different configurations of the synthetic datasets. We report the F1 and MCC scores for each method across different latent dimensions $D$. Average over 5 runs reported

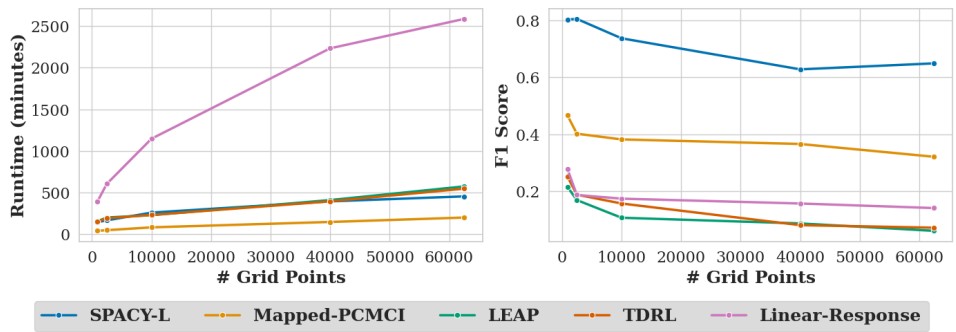

Figure 5: Comparison of runtime (in minutes) and F1 score across different grid sizes. The left plot shows how the runtime increases with grid size, while the right plot displays the corresponding F1 scores for causal discovery. Average over 5 runs reported.

datasets, Mapped PCMCI performs competitively, particularly when using linear spatial mapping, while LEAP, TDRL, and Linear-Response exhibit weaker performance. In the nonlinear settings, SPACY significantly outperforms the baselines, with a more pronounced performance drop observed for LEAP, TDRL, and Linear-Response, whose F1 scores decline sharply as $D$ increases. SPACY's performance scales more effectively with increasing $D$, further widening the gap in performance.

The quality of the causal representation, measured by the MCC score, follows a similar pattern. Mapped PCMCI remains competitive with SPACY, while LEAP, TDRL, and Linear-Response consistently show lower MCC scores across all configurations. Figure 10 provides a visual illustration of the recovered spatial factors.

**Scalability** We also measure the scalability of SPACY with increasing grid-size. For this experiment, we used the dataset with linear SCM and linear spatial mapping. Figure 5 demonstrates the scalability and performance of SPACY compared to the baseline methods as the grid size $L$ increases. The runtime plot indicates that, while all methods experience an increase in runtime with increasing grid size, SPACY strikes a good balance, exhibiting moderate growth in computational time while maintaining strong causal discovery performance. Although Mapped-PCMCI is the most efficient in terms of runtime, it underperforms in causal discovery. LEAP and TDRL show similar or higher computational costs than SPACY but fail to match its performance. Linear-Response, in particular, scales poorly in terms of runtime with increasing grid size.

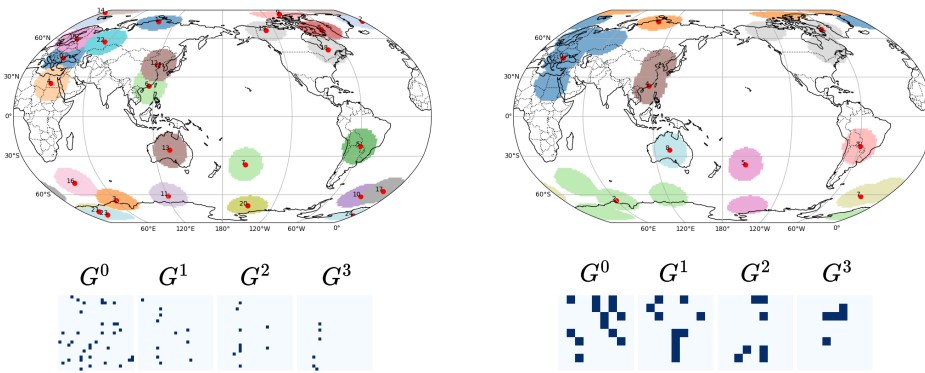

Figure 6: Visualization of (left) the learned spatial factors and causal graph (right) the learned spatial factors and causal graph after merging based on proximity and graph links.

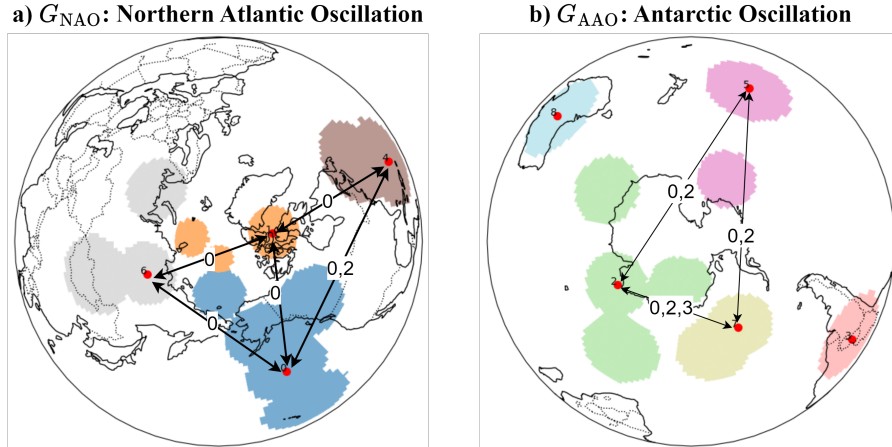

Figure 7: Qualitative results for Global Temperature climate dataset. The numbers on the arrow refers to the time lag of the causal links. Subgraph of G depicting learned causal relationships among regions associated with the (a) Northern Atlantic Oscillation (b) Antarctic Oscillation.

## 5.2 REAL-WORLD APPLICATION TO CLIMATE SCIENCE

The Global Temperature Dataset is a mixed real-simulated dataset containing monthly global temperature data from 1999 to 2001. It includes 7,531 simulated samples, each with a 24-month time sequence, across a $145 \times 192$ spatial grid. Before applying SPACY, we deseasonalized the data by subtracting the monthly mean values. Given the global nature of the dataset, we employed the Haversine distance instead of Euclidean distance when calculating the RBF kernel for spatial factors. For more details about the dataset and preprocessing steps, refer to Appendix A.6.

**Results.** We qualitatively evaluate SPACY's inferred spatial factors and causal graph due to the absence of a ground truth causal graph. Figure 6 illustrates the spatial factors and causal graphs learned by SPACY from the Global Temperature Dataset, visualized using the procedure outlined in Appendix A.3.3. The spatial modes identified by SPACY correspond to critical regions that significantly influence global climate patterns, including coastlines of major land masses (e.g., East Asia, Northern Europe) and key ocean areas (e.g., Central Pacific, South Atlantic)

Figure 7 highlights two subgraphs extracted from SPACY's results: $G_{\text{NAO}}$ and $G_{\text{AAO}}$, which correspond to spatial modes associated with the Northern Atlantic Oscillation (NAO) (Hurrell, 1995; Chen & den Dool, 2003; Hurrell et al., 2003) and the Antarctic Oscillation (AAO) (Thompson & Solomon, 2002; Mo, 2000). This subgraph reveals how SPACY uncovers causal connections between regions that share similar weather characteristics and are driven by these known teleconnec-

tion patterns. The model successfully identifies the spatial extent and connectivity of NAO-related regions, which comprises of North-Eastern Canada and North Western Europe (Chen & den Dool, 2003; Hurrell, 1995), and AAO-related regions (South-East Australia, South-Atlantic, South-Indian Ocean) (Thompson & Solomon, 2002). The learned subgraphs correctly mirror the correlation and oscillation of temperature in these regions, identifying both instantaneous links and those occurring a few months prior. Moreover, the inferred modes are spatially confined, each with a distinct center and scale, which enhances their interpretability. In contrast to standard principal component analyses and methods like Mapped PCMCI (Figure 13), which often result in broadly distributed components that are hard to interpret, SPACY infers localized regions with well-defined spatial extents.

## 5.3 ABLATION STUDIES

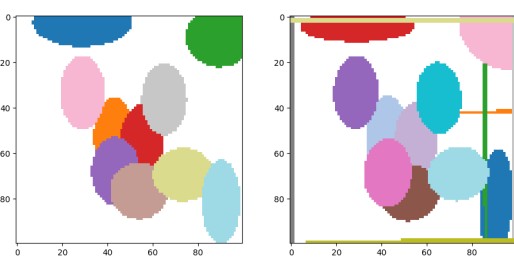

| $D^*$ | SPACY-L $(D = D^*)$ | SPACY-L $(D = D^* + 10)$ |
|---|---|---|
| 10 | $0.623 \pm 0.06$ | $\mathbf{0.642 \pm 0.07}$ |
| 20 | $\mathbf{0.752 \pm 0.03}$ | $0.549 \pm 0.03$ |
| 30 | $\mathbf{0.596 \pm 0.05}$ | $0.529 \pm 0.06$ |

(c) Causal discovery performance (F1-score)

(a) Ground truth modes          (b) Inferred modes

Figure 8: Overview of the results for over-specification ablation study. (a) Visualization of the ground-truth location and scale of the spatial modes. (b) Visualization of the inferred location and scale when we over-specify the number of nodes. (c) Causal discovery performance after matching and eliminating nodes. Average over 5 seeds reported

**Over-specifying $D$.** SPACY requires specifying the number of latent variables $D$ as a hyperparameter. In practice, the exact number of underlying factors is often unknown. We examine the effect of overspecifying $D$ by setting it to $D^* + 10$, where $D^*$ represents the true number of nodes used to generate the data. We use the synthetic dataset with grid dimensions $100 \times 100$, linear SCM and non-linear mapping.

Figure 8 illustrates the results of our experiment. When $D^* = 10$, despite over-specifying the number of nodes, the inferred spatial modes' general locations align well with the ground truth. The presence of additional modes does not significantly detract from the accuracy of detecting the primary spatial modes. This suggests that SPACY maintains robust learning of spatial representations even when $D$ exceeds the true number of spatial factors. This observation also holds true when comparing the causal discovery performance using the F1 score.

We also examine the robustness of SPACY to the choice of the kernel function when computing the spatial factors. The results are detailed in Appendix A.4.

## 6 CONCLUSION

In this work, we examined the problem of inferring causal relationships from spatiotemporal data. This problem has significant applications in climate, neuroscience, and biomedical science, among other fields. We proposed an end-to-end variational inference method to learn the latent causal representations and the underlying SCM, while producing an interpretable output. We discussed the structural identifiability of our model, and demonstrated the empirical efficacy of our method on both synthetic and simulated climate datasets. SPACY successfully recovers spatial patterns linked to known events like the Northern Atlantic Oscillation and Antarctic Oscillation.

As a direction for future work, our method can be extended to multivariate settings. Performing latent causal representation learning and causal discovery between multiple variables could further enhance the capability of our approach in handling complex real-world datasets. Such an extension would be particularly valuable in domains like climate science (Tibau, 2022; Brouillard et al., 2024), where interactions between multiple variables (e.g. temperature and pressure) are critical.

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

# A APPENDIX

## A.1 THEORY

### A.1.1 ELBO DERIVATION

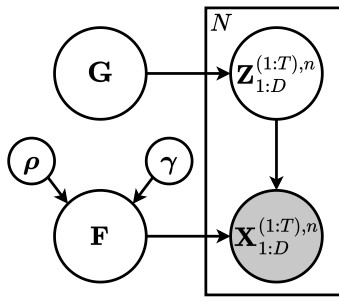

$$\mathbf{Z}_d^{(t)} = f_d\left(\mathrm{Pa}_G^d(<t), \mathrm{Pa}_G^d(t)\right) + \eta_d^{(t)}$$
$$\boldsymbol{\rho}_d \sim U[0,1]^K, \boldsymbol{\gamma}_d \sim \mathcal{N}(0, I)$$
$$\mathbf{F}_d = [\mathrm{RBF}_d(x_\ell; \boldsymbol{\rho}_d, \boldsymbol{\gamma}_d)]_{\ell=1}^L, \quad x_\ell \in \mathcal{G}$$
$$\mathbf{X}_\ell = g_\ell([\mathbf{FZ}]_\ell) + \varepsilon_\ell$$
$$\varepsilon_\ell \sim \mathcal{N}(0, \sigma_\ell^2 I)$$

Figure 9: Probabilistic graphical model for SPACY and the generative equations. Shaded circles are observed and hollow circles are latent.

**Proposition 1.** *The data generation model described in Figure 2 admits the following evidence lower bound (ELBO):*

$$\log p_\theta\left(\mathbf{X}^{(1:T),1:N}\right) \geq \sum_{n=1}^N \left\{ \mathbb{E}_{q_\phi(\mathbf{Z}^{(1:T),n}|\mathbf{X}^{(1:T),n})q_\phi(\mathbf{G})q_\phi(\mathbf{F})}\left[\log p_\theta\left(\mathbf{X}^{(1:T),n}|\mathbf{Z}^{(1:T),n}, \mathbf{F}\right)\right.\right.$$

$$\left.\left.+ \left[\log p_\theta\left(\mathbf{Z}^{(1:T),n}|\mathbf{G}\right) - \log q_\phi\left(\mathbf{Z}^{(1:T),n}|\mathbf{X}^{(1:T),n}\right)\right]\right]\right\} + \mathbb{E}_{q_\phi(\mathbf{G})}[\log p(\mathbf{G}) - \log q_\phi(\mathbf{G})]$$

$$+ \mathbb{E}_{q_\phi(\mathbf{F})}[\log p(\mathbf{F}) - \log q_\phi(\mathbf{F})] = \text{ELBO}(\theta, \phi)$$

*Proof.* We begin with the log-likelihood of the observed data:

$$\log p_\theta\left(\mathbf{X}^{(1:T),1:N}\right) = \log \int p_\theta\left(\mathbf{X}^{(1:T),1:N}, \mathbf{Z}^{(1:T),1:N}, \mathbf{G}, \mathbf{F}\right) d\mathbf{Z}\, d\mathbf{G}\, d\mathbf{F}$$

We multiply and divide by the variational distribution $q_\phi\left(\mathbf{Z}^{(1:T),1:N}|\mathbf{X}^{(1:T),1:N}\right) q_\phi(\mathbf{G})\, q_\phi(\mathbf{F})$ to create an evidence lower bound (ELBO) using Jensen's inequality:

$$\log p_\theta\left(\mathbf{X}^{(1:T),1:N}\right)$$

$$= \log \int \frac{q_\phi\left(\mathbf{Z}^{(1:T),1:N}|\mathbf{X}^{(1:T),1:N}\right) q_\phi(\mathbf{G})\, q_\phi(\mathbf{F})}{q_\phi\left(\mathbf{Z}^{(1:T),1:N}|\mathbf{X}^{(1:T),1:N}\right) q_\phi(\mathbf{G})\, q_\phi(\mathbf{F})} p_\theta\left(\mathbf{X}^{(1:T),1:N}, \mathbf{Z}^{(1:T),1:N}, \mathbf{G}, \mathbf{F}\right) d\mathbf{Z}\, d\mathbf{G}\, d\mathbf{F}$$

$$\geq \mathbb{E}_{q_\phi(\mathbf{Z}^{(1:T),1:N}|\mathbf{X}^{(1:T),1:N})q_\phi(\mathbf{G})q_\phi(\mathbf{F})}\left[\log \frac{p_\theta\left(\mathbf{X}^{(1:T),1:N}, \mathbf{Z}^{(1:T),1:N}, \mathbf{G}, \mathbf{F}\right)}{q_\phi\left(\mathbf{Z}^{(1:T),1:N}|\mathbf{X}^{(1:T),1:N}\right) q_\phi(\mathbf{G})\, q_\phi(\mathbf{F})}\right]. \tag{9}$$

By the assumptions of the data generative process,

$$p_\theta\left(\mathbf{X}^{(1:T),1:N}, \mathbf{Z}^{(1:T),1:N}, \mathbf{G}, \mathbf{F}\right) = p_\theta\left(\mathbf{X}^{(1:T),1:N}|\mathbf{Z}^{(1:T),1:N}, \mathbf{F}\right) p_\theta\left(\mathbf{Z}^{(1:T),1:N}|\mathbf{G}\right) p\left(\mathbf{F}\right) p\left(\mathbf{G}\right)$$

Further, note that $\mathbf{X}^{(1:T),1:N}$ are conditionally independent given $\mathbf{F}, \mathbf{Z}^{(1:T),1:N}$. Also, $\mathbf{X}^{(1:T),n}$ is conditionally independent of $\mathbf{Z}^{(1:T),m}$ given $\mathbf{Z}^{(1:T),n}, \mathbf{F}$ for $m \neq n$. This implies that:

$$p_\theta\left(\mathbf{X}^{(1:T),1:N}|\mathbf{Z}^{(1:T),1:N}, \mathbf{F}\right) = \prod_{n=1}^{N} p_\theta\left(\mathbf{X}^{(1:T),n}|\mathbf{Z}^{(1:T),n}, \mathbf{F}\right).$$

Similarly, $\mathbf{Z}^{(1:T),1:N}$ are conditionally independent given $\mathbf{G}$, which implies

$$p_\theta\left(\mathbf{Z}^{(1:T),1:N}|\mathbf{G}\right) = \prod_{n=1}^{N} p_\theta\left(\mathbf{Z}^{(1:T),n}|\mathbf{G}\right).$$

Substituting these terms back into equation 9 and grouping terms according to the variables $\mathbf{Z}, \mathbf{G}, \mathbf{F}$ yields the ELBO.

$$\begin{aligned}
\log p_\theta\left(\mathbf{X}^{(1:T),1:N}\right) \geq \sum_{n=1}^{N} &\left\{ \mathbb{E}_{q_\phi(\mathbf{Z}^{(1:T),n}|\mathbf{X}^{(1:T),n})q_\phi(\mathbf{G})q_\phi(\mathbf{F})} \left[ \log p_\theta\left(\mathbf{X}^{(1:T),n}|\mathbf{Z}^{(1:T),n}, \mathbf{G}, \mathbf{F}\right) \right.\right. \\
&\left.\left. + \left( \log p_\theta\left(\mathbf{Z}^{(1:T),n}|\mathbf{G}\right) - \log q_\phi\left(\mathbf{Z}^{(1:T),n}|\mathbf{X}^{(1:T),n}\right) \right) \right] \right\} \\
&+ \mathbb{E}_{q_\phi(\mathbf{G})} \left[ \log p(\mathbf{G}) - \log q_\phi(\mathbf{G}) \right] \\
&+ \mathbb{E}_{q_\phi(\mathbf{F})} \left[ \log p(\mathbf{F}) - \log q_\phi(\mathbf{F}) \right] \equiv \text{ELBO}(\theta, \phi).
\end{aligned}$$

$\square$

### A.1.2 IDENTIFIABILITY

**Definition 1** (Linearly Independent Family). Let $\mathcal{F}$ be a family of real-valued, parametric functions $\mathcal{F} = \left\{ f_\psi : \mathbb{R}^K \to \mathbb{R} \right\}$. $\mathcal{F}$ is said to be a linearly independent family if, for any finite set $\{\psi_1, ..., \psi_n\}$, we have

$$\sum_{k=1}^{n} \alpha_k f_{\psi_k} = 0 \implies \alpha_k = 0 \quad \forall k \in [n]. \tag{10}$$

**Definition 2** (Spatial Factor Process). Let $\mathcal{G} = [0, 1]^K$ be a $K-$dimensional grid, and let $\mathbf{Z} \in \mathbb{R}^{D \times T}$. Suppose $\mathcal{F} = \{F_{\psi_1}, ..., F_{\psi_D}\}$ is a finite linearly independent family. We define a Spatial Factor Process SFP$(\mathbf{Z}, \mathcal{F}, p_\varepsilon)$, denoted by $\mathbf{X} : \mathcal{G} \to \mathbb{R}^T$, as follows:

For each location $x \in \mathcal{G}$ in the grid,

$$\mathbf{X}(x) = \mathbf{F}_x^\top \mathbf{Z} + \varepsilon_x \tag{11}$$

where

$$\mathbf{F}_x = \begin{bmatrix} F_{\psi_1}(x) \\ \vdots \\ F_{\psi_D}(x) \end{bmatrix},$$

and $\varepsilon_x \sim p_\varepsilon(\cdot)$ is a normally distributed noise term

A Spatial Factor Process (SFP) extends the concept of a gridded time series to an infinite resolution. Instead of observing the time series on a finite set of grid points, we assume that a time series can be observed at every location within a bounded $K$-dimensional grid, $\mathcal{G} = [0, 1]^K$. In the above definition, $\mathbf{Z}$ represents a (fixed) realization of a $D-$dimensional time series of length $T$.

We now show that SFPs are identifiable, i.e., if the distributions of two SFPs are equal, then their corresponding parameters $\mathbf{Z}$ and $\mathbf{F}$ are also equal (upto permutation).

**Theorem 3** (Identifiability of SFPs). *If we have two SFPs $\mathbf{X} = SFP(\mathbf{Z}, \mathcal{F}, p_\varepsilon)$ and $\mathbf{Y} = SFP(\tilde{\mathbf{Z}}, \tilde{\mathcal{F}}, q_\varepsilon)$ where none of the rows of $\mathbf{Z}$ or $\tilde{\mathbf{Z}}$ are all zero, such that $p(\mathbf{X}(x)) = p(\mathbf{Y}(x))$ for every $x \in \mathcal{G}$, then $\mathbf{Z} = P\tilde{\mathbf{Z}}$ and $\mathcal{F} = \tilde{\mathcal{F}}$ for some permutation matrix $P$.*

*Proof.* Note that, for every $\mathbf{v} \in \mathbb{R}^T$,

$$p(\mathbf{X}(x) = \mathbf{v}) = p(\mathbf{Y}(x) = \mathbf{v})$$
$$\implies p_\varepsilon\left(\varepsilon_x = \mathbf{v} - \mathbf{y}\right) = q_\varepsilon\left(\tilde{\varepsilon}_x = \mathbf{v} - \tilde{\mathbf{y}}\right)$$

where $\mathbf{y} = \mathbf{F}_x^\top \mathbf{Z}$ and $\tilde{\mathbf{y}} = \tilde{\mathbf{F}}_x^\top \tilde{\mathbf{Z}}$. Since $p_\varepsilon$ and $q_\varepsilon$ are normally distributed, this can only be true when

$$\mathbf{y} = \tilde{\mathbf{y}}$$
$$\implies \mathbf{F}_x^\top \mathbf{Z} = \tilde{\mathbf{F}}_x^\top \tilde{\mathbf{Z}} \quad \forall x \in \mathcal{G}$$
$$\implies \sum_{j=1}^{D} F_{\psi_j}(x) z_{jt} = \sum_{j=1}^{D} F_{\tilde{\psi}_j}(x) \tilde{z}_{jt} \quad \forall x \in \mathcal{G}, t \in [T]$$
$$\implies \sum_{j=1}^{D} F_{\psi_j}(x) z_{jt} - \sum_{j=1}^{D} F_{\tilde{\psi}_j}(x) \tilde{z}_{jt} = 0 \quad \forall x \in \mathcal{G}, t \in [T] \tag{12}$$

Suppose $\{\psi_1, \ldots, \psi_D\} \cap \{\tilde{\psi}_1, \ldots, \tilde{\psi}_D\} = \varnothing$. Then, this would imply that $z_{jt} = \tilde{z}_{jt} = 0 \quad \forall j, t$, which is a contradiction since we assume that none of the time series are all 0. This implies that $\{\psi_1, \ldots, \psi_D\} \cap \{\tilde{\psi}_1, \ldots, \tilde{\psi}_D\} \neq \varnothing$. Assume $V = \left\{ (i, j) : \psi_i = \tilde{\psi}_j \right\}$ and define $I = \{i : \exists j \text{ such that } (i, j) \in V\}$, $J = \{j : \exists i \text{ such that } (i, j) \in V\}$. Define the function $\mathcal{V} : I \to J$, $\mathcal{V}(i) = j$ such that $(i, j) \in V$. Then equation 12 can be written as:

$$\sum_{\substack{j=1 \\ j \notin I}}^{D} F_{\psi_j}(x) z_{jt} - \sum_{\substack{j=1 \\ j \notin J}}^{D} F_{\tilde{\psi}_j}(x) \tilde{z}_{jt} + \sum_{\substack{j=1 \\ j \in I}}^{D} F_{\psi_j}(x) \left( z_{jt} - \tilde{z}_{\mathcal{V}(j)t} \right) = 0 \quad \forall x \in \mathcal{G}, t \in [T].$$

If $I \neq \varnothing$, then $z_{jt} = 0 \ \forall j \notin I$ due to the linear independence of $F_{\psi_j}$, which contradicts our assumption of non-zero time series. Therefore, we must have that $\{\psi_1, \ldots, \psi_D\} = \{\tilde{\psi}_1, \ldots, \tilde{\psi}_D\}$, and $z_{jt} = \tilde{z}_{\mathcal{V}(j)t} \ \forall j, t$. $\qquad\square$

We now consider the identifiability of the parameters from the observational distribution. To this end, we first introduce a useful lemma. We adapt the arguments from Lemma 3 in Boussard et al. (2023) with some modifications.

**Lemma 1** (Denoising $\mathbf{X}$). *Assume we have two models $\mathbf{X}$ and $\tilde{\mathbf{X}}$ with spatial factors $\mathbf{F}_x$ and $\tilde{\mathbf{F}}_x$ respectively. Assume that the observational distributions of $\mathbf{X}(x)$ and $\tilde{\mathbf{X}}(x)$ are equal, i.e., the following property holds:*

*For any finite set of grid points $\{x_1, \ldots, x_n\} \in \mathcal{G}$, we have*

$$p\left(\mathbf{X}(x_1) = \chi_1, \ldots, \mathbf{X}(x_n) = \chi_n\right) = p\left(\tilde{\mathbf{X}}(x_1) = \chi_1, \ldots, \tilde{\mathbf{X}}(x_n) = \chi_n\right) \tag{13}$$

*for all values of $(\chi_1, \ldots, \chi_n) \in \mathbb{R}^T \times \mathbb{R}^n$. Then we have that the following holds:*

*Given any set of points $\{x_1', \ldots, x_k'\}$, we have that $p\left(\mathbf{Y}(x_1'), \ldots, \mathbf{Y}(x_k')\right) = p\left(\tilde{\mathbf{Y}}(x_1'), \ldots, \tilde{\mathbf{Y}}(x_k')\right)$, where $\mathbf{Y}(x) := \mathbf{F}_x^\top \mathbf{Z}$ and $\tilde{\mathbf{Y}}(x) := \tilde{\mathbf{F}}_x^\top \tilde{\mathbf{Z}}$.*

*Proof.* Pick $n$ distinct grid points $\{x_1, \ldots, x_n\} \subseteq \mathcal{G}$ such that $\{x_1', \ldots, x_k'\} \cap \{x_1, \ldots, x_n\} = \phi$ and $n + k > D$. Let $L = n + k$. Then, we can use the same argument as in Lemma 3 in Boussard et al. (2023) on the distributions of $\{\mathbf{X}(x_1), \ldots, \mathbf{X}(x_n), \mathbf{X}(x_1'), \ldots, \mathbf{X}(x_k')\}$ and $\left\{\tilde{\mathbf{X}}(x_1), \ldots, \tilde{\mathbf{X}}(x_n), \tilde{\mathbf{X}}(x_1'), \ldots, \tilde{\mathbf{X}}(x_k')\right\}$, which we repeat for the sake of completeness.

Let $\mathbb{P}_{\mathbf{X}(x_1),\ldots,\mathbf{X}(x_n),\mathbf{X}(x_1'),\ldots,\mathbf{X}(x_k')}$ and $\mathbb{P}_{\tilde{\mathbf{X}}(x_1),\ldots,\tilde{\mathbf{X}}(x_n),\tilde{\mathbf{X}}(x_1'),\ldots,\tilde{\mathbf{X}}(x_k')}$ denote the probability measures corresponding to the densities

$$p(\mathbf{X}(x_1),\ldots,\mathbf{X}(x_n),\mathbf{X}(x_1'),\ldots,\mathbf{X}(x_k')) := \int p(\mathbf{X}(x_1),\ldots,\mathbf{X}(x_n),\mathbf{X}(x_1'),\ldots,\mathbf{X}(x_k'),\mathbf{Z})\,d\mathbf{Z},$$

$$p(\tilde{\mathbf{X}}(x_1),\ldots,\tilde{\mathbf{X}}(x_n),\tilde{\mathbf{X}}(x_1'),\ldots,\tilde{\mathbf{X}}(x_k')) := \int p(\tilde{\mathbf{X}}(x_1),\ldots,\tilde{\mathbf{X}}(x_n),\tilde{\mathbf{X}}(x_1'),\ldots,\tilde{\mathbf{X}}(x_k'),\tilde{\mathbf{Z}})\,d\tilde{\mathbf{Z}},$$

respectively.

It is given that:

$$\mathbb{P}_{\mathbf{X}(x_1),\ldots,\mathbf{X}(x_n),\mathbf{X}(x_1'),\ldots,\mathbf{X}(x_k')} = \mathbb{P}_{\tilde{\mathbf{X}}(x_1),\ldots,\tilde{\mathbf{X}}(x_n),\tilde{\mathbf{X}}(x_1'),\ldots,\tilde{\mathbf{X}}(x_k')}$$

Define $\mathbf{Y}(x) := \mathbf{F}_x^\top \mathbf{Z}$ and $\tilde{\mathbf{Y}}(x) := \tilde{\mathbf{F}}_x^\top \tilde{\mathbf{Z}}$, where $\mathbf{Z} \sim p(\mathbf{Z})$ and $\tilde{\mathbf{Z}} \sim p(\tilde{\mathbf{Z}})$. Let $\mathcal{Y} = (\mathbf{Y}(x_1),\ldots,\mathbf{Y}(x_n),\mathbf{Y}(x_1'),\ldots,\mathbf{Y}(x_k'))$ and $\tilde{\mathcal{Y}} = \left(\tilde{\mathbf{Y}}(x_1),\ldots,\tilde{\mathbf{Y}}(x_n),\tilde{\mathbf{Y}}(x_1'),\ldots,\tilde{\mathbf{Y}}(x_k')\right)$. Let $\mathbb{P}_{\mathbf{Y}(x)}$ and $\mathbb{P}_{\tilde{\mathbf{Y}}(x)}$ be the distributions of $\mathbf{Y}(x)$ and $\tilde{\mathbf{Y}}(x)$, respectively. We have:

$$\mathbf{X}(x) = \mathbf{Y}(x) + \varepsilon_x, \quad \tilde{\mathbf{X}}(x) = \tilde{\mathbf{Y}}(x) + \tilde{\varepsilon}_x,$$

where $\varepsilon_x \sim \mathcal{N}(0, \sigma^2 I_T)$ and $\tilde{\varepsilon}_x \sim \mathcal{N}(0, \tilde{\sigma}^2 I_T)$.

Denote $\boldsymbol{\varepsilon} = \left(\varepsilon_{x_1},\ldots,\varepsilon_{x_n},\varepsilon_{x_1'},\ldots,\varepsilon_{x_k'}\right)$ and $\tilde{\boldsymbol{\varepsilon}} = \left(\tilde{\varepsilon}_{x_1},\ldots,\tilde{\varepsilon}_{x_n},\tilde{\varepsilon}_{x_1'},\ldots,\tilde{\varepsilon}_{x_k'}\right)$.

By the additive structure of the model, the equality of measures becomes a convolution equation:

$$\mathbb{P}_{\mathcal{Y}} * \mathbb{P}_{\boldsymbol{\varepsilon}} = \mathbb{P}_{\tilde{\mathcal{Y}}} * \mathbb{P}_{\tilde{\boldsymbol{\varepsilon}}},$$

where $\mathbb{P}_{\boldsymbol{\varepsilon}}$ and $\mathbb{P}_{\tilde{\boldsymbol{\varepsilon}}}$ represent the measures of the Gaussian noise terms, and $*$ denotes convolution.

Applying the Fourier transform $\mathscr{F}$ to both sides and using the fact that the Fourier transform of a convolution is the product of the Fourier transforms (Pollard, 2002),

$$\mathscr{F}\left(\mathbb{P}_{\mathcal{Y}} * \mathbb{P}_{\boldsymbol{\varepsilon}}\right) = \mathscr{F}\left(\mathbb{P}_{\tilde{\mathcal{Y}}} * \mathbb{P}_{\tilde{\boldsymbol{\varepsilon}}}\right)$$

$$\implies \mathscr{F}\left(\mathbb{P}_{\mathcal{Y}}\right)\mathscr{F}\left(\mathbb{P}_{\boldsymbol{\varepsilon}}\right) = \mathscr{F}\left(\mathbb{P}_{\tilde{\mathcal{Y}}}\right)\mathscr{F}\left(\mathbb{P}_{\tilde{\boldsymbol{\varepsilon}}}\right).$$

Given that the Fourier transform of a zero-mean Gaussian random vector with covariance $\sigma^2 I_{LT}$ is $e^{-\frac{\sigma^2}{2}\omega^\top \omega}$, we can rewrite the above as:

$$\mathscr{F}\left(\mathbb{P}_{\mathbf{Y}(x)}\right)(\omega)e^{-\frac{\sigma^2}{2}\omega^\top \omega} = \mathscr{F}\left(\mathbb{P}_{\tilde{\mathbf{Y}}(x)}\right)(\omega)e^{-\frac{\tilde{\sigma}^2}{2}\omega^\top \omega}, \quad \forall \omega \in \mathbb{R}^T.$$

We now aim to show that $\sigma^2 = \tilde{\sigma}^2$. Assume, without loss of generality, that $\sigma^2 < \tilde{\sigma}^2$. Dividing both sides by $e^{-\frac{\sigma^2}{2}\omega^\top \omega}$ yields:

$$\mathscr{F}\left(\mathbb{P}_{\mathcal{Y}}\right)(\omega) = \mathscr{F}\left(\mathbb{P}_{\tilde{\mathcal{Y}}}\right)(\omega)e^{-\frac{\tilde{\sigma}^2-\sigma^2}{2}\omega^\top \omega}, \quad \forall \omega \in \mathbb{R}^{LT}.$$

Here, $e^{-\frac{\tilde{\sigma}^2-\sigma^2}{2}\omega^\top \omega}$ is the Fourier transform of a Gaussian distribution with covariance $(\tilde{\sigma}^2-\sigma^2)I_{LT}$. However, note that the left-hand side is the Fourier transform of a distribution supported on the column span of $\mathbf{F}_x$, which lies in a $D$-dimensional subspace of $\mathbb{R}^{LT}$. In contrast, the right-hand side corresponds to a distribution with full support in $\mathbb{R}^{LT}$, as it involves the convolution of $\mathbb{P}_{\tilde{\mathcal{Y}}}$ with a $LT$-dimensional Gaussian random variable. This is a contradiction, as the supports of the distributions on both sides must match.

Thus, we must have $\sigma^2 = \tilde{\sigma}^2$.

Finally, with $\sigma^2 = \tilde{\sigma}^2$, we conclude that:

$$\mathscr{F}\left(\mathbb{P}_{\mathcal{Y}}\right) = \mathscr{F}\left(\mathbb{P}_{\tilde{\mathcal{Y}}}\right),$$

$$\mathbb{P}_{\mathcal{Y}} = \mathbb{P}_{\tilde{\mathcal{Y}}}.$$

Marginalizing out the variables $\mathbf{Y}(x_1),\ldots,\mathbf{Y}(x_n)$ and $\tilde{\mathbf{Y}}(x_1),\ldots,\tilde{\mathbf{Y}}(x_n)$ yields the desired result.

$\square$

**Theorem 4** (Identifiability of the latents). *Suppose we have two spatial factor processes $\mathbf{X}(x)$ and $\tilde{\mathbf{X}}(x)$ with spatial factors $\mathbf{F}_x$ and $\tilde{\mathbf{F}}_x$ respectively, generated from linearly independent families $\mathcal{F} = \{f_{\psi_1}, \ldots, f_{\psi_D}\}$ and $\tilde{\mathcal{F}} = \left\{f_{\tilde{\psi}_1}, \ldots, f_{\tilde{\psi}_D}\right\}$ respectively.*

*Suppose the observational distributions of $\mathbf{X}(x)$ and $\tilde{\mathbf{X}}(x)$ are equal, i.e., the following property holds:*

*For any finite set of grid points $\{x_1, \ldots, x_n\} \in \mathcal{G}$, we have*

$$p\left(\mathbf{X}(x_1) = \chi_1, \ldots, \mathbf{X}(x_n) = \chi_n\right) = p\left(\tilde{\mathbf{X}}(x_1) = \chi_1, \ldots, \tilde{\mathbf{X}}(x_n) = \chi_n\right) \tag{14}$$

*for all values of $(\chi_1, \ldots, \chi_n) \in \mathbb{R}^T \times \mathbb{R}^n$.*

*Then the latent variable distribution is identifiable upto transformation by an invertible matrix.*

*Proof.* Since equation 14 holds, we can apply Lemma 1, by which we have that:

$$p\left(\mathbf{Y}(x_1) = \mathbf{y}_1, \ldots, \mathbf{Y}(x_n) = \mathbf{y}_n\right) = p\left(\tilde{\mathbf{Y}}(x_1) = \mathbf{y}_1, \ldots, \tilde{\mathbf{Y}}(x_n) = \mathbf{y}_n\right) \tag{15}$$

$\forall (\mathbf{y}_1, \ldots, \mathbf{y}_n) \in \mathbb{R}^T \times \mathbb{R}^n$ where $\mathbf{Y}(x) = \mathbf{F}_x^\top \mathbf{Z}$.

Since the family $\mathcal{F}$ is linearly independent, we can pick $D$ points $\{x_1, \ldots, x_D\}$ from $\mathcal{G}$ such that

$$\mathfrak{F} = \begin{bmatrix} f_{\psi_1}(x_1) & \cdots & f_{\psi_D}(x_1) \\ \vdots & & \vdots \\ f_{\psi_1}(x_D) & \cdots & f_{\psi_D}(x_D) \end{bmatrix} = \begin{bmatrix} \underline{\quad} & \mathbf{F}_{x_1}^\top & \underline{\quad} \\ \vdots & \vdots & \vdots \\ \underline{\quad} & \mathbf{F}_{x_D}^\top & \underline{\quad} \end{bmatrix}$$

is full rank [1].

Similarly, we can pick $D$ points $\{\tilde{x}_1, \ldots, \tilde{x}_D\}$ from $\mathcal{G}$ such that

$$\tilde{\mathfrak{F}} = \begin{bmatrix} f_{\tilde{\psi}_1}(\tilde{x}_1) & \cdots & f_{\tilde{\psi}_D}(\tilde{x}_1) \\ \vdots & & \vdots \\ f_{\tilde{\psi}_1}(\tilde{x}_D) & \cdots & f_{\tilde{\psi}_D}(\tilde{x}_D) \end{bmatrix} = \begin{bmatrix} \underline{\quad} & \tilde{\mathbf{F}}_{\tilde{x}_1}^\top & \underline{\quad} \\ \vdots & \vdots & \vdots \\ \underline{\quad} & \tilde{\mathbf{F}}_{\tilde{x}_D}^\top & \underline{\quad} \end{bmatrix}$$

is full rank.

Define:

$$\mathcal{Y} = \begin{bmatrix} \mathbf{Y}(x_1) \\ \vdots \\ \mathbf{Y}(x_D) \end{bmatrix}, \quad \tilde{\mathcal{Y}} = \begin{bmatrix} \tilde{\mathbf{Y}}(x_1) \\ \vdots \\ \tilde{\mathbf{Y}}(x_D) \end{bmatrix}.$$

Let $\mathbf{y} = (\mathbf{y}_1, \ldots, \mathbf{y}_D)$ and $\tilde{\mathbf{y}} = (\tilde{\mathbf{y}}_1, \ldots, \tilde{\mathbf{y}}_D)$.

Observe that

$$\mathcal{Y} = \mathfrak{F}\mathbf{Z}$$
$$\tilde{\mathcal{Y}} = \tilde{\mathfrak{F}}\tilde{\mathbf{Z}}$$

Using the formula for transformation of random variables,

$$p\left(\mathcal{Y} = \mathbf{y}\right) = \left|\det\left(\mathfrak{F}\right)\right| p\left(\mathbf{Z} = \mathfrak{F}^{-1}\mathbf{y}\right), \forall \mathbf{y} \in \mathbb{R}^{D \times T}$$
$$p\left(\tilde{\mathcal{Y}} = \mathbf{y}\right) = \left|\det\left(\tilde{\mathfrak{F}}\right)\right| p\left(\tilde{\mathbf{Z}} = \tilde{\mathfrak{F}}^{-1}\mathbf{y}\right), \forall \mathbf{y} \in \mathbb{R}^{D \times T}$$

---

[1] See for example `https://math.stackexchange.com/questions/3516189/prove-existence-of-evaluation-points-such-that-the-matrix-has-nonzero-determinan`

Applying equation 15 for the points $\{x_1, \ldots, x_D\}$, we can obtain

$$|\det(\mathfrak{F})| \, p\left(\mathbf{Z} = \mathfrak{F}^{-1}\mathbf{y}\right) = \left|\det\left(\tilde{\mathfrak{F}}\right)\right| p\left(\tilde{\mathbf{Z}} = \tilde{\mathfrak{F}}^{-1}\mathbf{y}\right), \, \forall \mathbf{y} \in \mathbb{R}^{D \times T}$$

$$\implies p\left(\mathbf{Z} = \mathfrak{F}^{-1}\mathbf{y}\right) = \frac{\left|\det\left(\tilde{\mathfrak{F}}\right)\right|}{|\det(\mathfrak{F})|} \times p\left(\tilde{\mathbf{Z}} = \tilde{\mathfrak{F}}^{-1}\mathbf{y}\right), \, \forall \mathbf{y} \in \mathbb{R}^{D \times T}.$$

Making the substitution $\mathbf{z} = \mathfrak{F}^{-1}\mathbf{y}$ and writing $\mathcal{M} = \tilde{\mathfrak{F}}^{-1}\mathfrak{F}$ yields:

$$p\left(\mathbf{Z} = z\right) = \frac{\left|\det\left(\tilde{\mathfrak{F}}\right)\right|}{|\det(\mathfrak{F})|} \times p\left(\tilde{\mathbf{Z}} = \mathcal{M}z\right), \, \forall \mathbf{y} \in \mathbb{R}^{D \times T}.$$

for the invertible matrix $\mathcal{M}$. Thus, we can recover the latent distribution up to transformation via an invertible matrix. $\qquad\square$

## A.2 IMPLEMENTATION DETAILS

### A.2.1 LOSS TERMS

We explain how we implement the various loss terms in equation 7.

The first term $\log p_\theta\left(\mathbf{X}^{(1:T),n}|\mathbf{Z}^{(1:T),n}, \mathbf{F}\right)$ in equation 7 represents the conditional likelihood of the observed data $\mathbf{X}^{(1:T),n}$ conditioned on $\mathbf{Z}^{(1:T),n}$ and $\mathbf{F}$. This is calculated as the mean squared error (MSE) between the recovered and original time series:

$$\log p_\theta\left(\mathbf{X}^{(1:T),n}|\mathbf{Z}^{(1:T),n}, \mathbf{F}\right) = \sum_{\ell=1}^{L} \left\|\mathbf{X}_\ell^{(1:T),n} - \widehat{\mathbf{X}}_\ell^{(1:T),n}\right\|^2$$

where $\widehat{\mathbf{X}}_\ell^{(t),n} = g_\ell\left([\mathbf{F}\mathbf{Z}]_\ell^{(t)}\right)$ is the reconstructed time-series from the spatial factor $\mathbf{F}$ and latent time series $\mathbf{Z}$ sampled from the variational distributions.

The term $\log p_\theta\left(\mathbf{Z}^{(1:T),n}|\mathbf{G}\right)$ denotes the conditional likelihood of the latent time-series given the sampled graph $\mathbf{G}$.

For SPACY-L, this is implemented as follows:

$$\log p_\theta\left(\mathbf{Z}^{(1:T),n}\Big|\mathbf{G}\right) = \sum_{t=L}^{T}\sum_{d=1}^{D} \log p_\theta\left(\mathbf{Z}_d^{(t),n}\Big|\mathrm{Pa}_\mathbf{G}^d(\leq t)\right)$$

$$= \sum_{t=L}^{T}\sum_{d=1}^{D} \left[\mathbf{Z}_d^{(t),n} - \sum_{k=0}^{\tau}\sum_{j=1}^{D} (\mathbf{G} \circ W)_{j,d}^k \times \mathbf{Z}_j^{(t-k),n}\right]^2.$$

For SPACY-NL, the equation follows from the conditional spline flow model employed in Durkan et al. (2019); Gong Wenbo & Nick (2022). The conditional spline flow model handles more flexible noise distributions, and can also model history-dependent noise. The structural equations are modeled as follows:

$$\mathbf{Z}_d^{(t)} = f_d\left(\mathrm{Pa}_\mathbf{G}^d(< t), \mathrm{Pa}_\mathbf{G}^d(t)\right) + w_d\left(\mathrm{Pa}_\mathbf{G}^d(< t)\right),$$

where $f_d\left(\mathrm{Pa}_\mathbf{G}^d(< t), \mathrm{Pa}_\mathbf{G}^d(t)\right)$ takes the form presented in equation 2. The spline flow model uses hypernetwork that predicts parameters for the conditional spline flow model, with embeddings $\mathscr{E}$, and hypernetworks $\xi_\eta$ and $\lambda_\eta$. The only difference is that the output dimension of $\xi_\eta$ is different, being equal to the number of spline parameters.

The noise variables $\eta_d^{(t)}$ are described using a conditional spline flow model,

$$p_{w_d}(w_d(\eta_d^{(t)}) \mid \mathrm{Pa}_\mathbf{G}^d(< t)) = p_\eta(\eta_d^{(t)}) \left|\frac{\partial(w_d)^{-1}}{\partial \eta_d^{(t)}}\right|, \tag{16}$$

with $\eta_d^{(t)}$ modeled as independent Gaussian noise.

The marginal likelihood becomes:

$$\log p_\theta \left( \mathbf{Z}^{(1:T),n} \middle| \mathbf{G} \right) = \sum_{t=\tau}^{T} \sum_{d=1}^{D} \log p_\theta \left( \mathbf{Z}_d^{(t),n} \middle| \mathrm{Pa}_\mathbf{G}^d(<t), \mathrm{Pa}_\mathbf{G}^d(t) \right)$$

$$= \sum_{t=\tau}^{T} \sum_{d=1}^{D} \log p_{w_d} \left( u_d^{(t),n} \middle| \mathrm{Pa}_\mathbf{G}^d(<t) \right) \quad (17)$$

where $u_d^{(t),n} = \mathbf{Z}_d^{(t),n} - f_d \left( \mathrm{Pa}_\mathbf{G}^d(<t), \mathrm{Pa}_\mathbf{G}^d(t) \right)$.

The prior distribution $p(\mathbf{G})$ is modeled as follows:

$$p(\mathbf{G}) \propto \exp \left( -\alpha \left\| \mathbf{G}^{(0:T)} \right\|^2 - \sigma h \left( \mathbf{G}^0 \right) \right). \quad (18)$$

The first term is a sparsity prior and $h \left( \mathbf{G}_0 \right)$ is the acyclicity constraint from (Zheng et al., 2018).

The terms $\mathbb{E}_{q_\phi(\mathbf{Z}^{(1:T),n}|\mathbf{X}^{(1:T),n})} \left[ -\log q_\phi \left( \mathbf{Z}^{(1:T),n}|\mathbf{X}^{(1:T),n} \right) \right], \mathbb{E}_{q_\phi(\mathbf{G})}[-\log q_\phi(\mathbf{G})]$ and $\mathbb{E}_{q_\phi(\mathbf{F})}[-\log q_\phi(\mathbf{F})]$ represent the entropies of the variational distributions and are evaluated in closed form, since their parameters are modeled as samples from Gaussian and Bernoulli distributions.

Finally, the prior term $p(F)$ is evaluated based on the assumed generative distribution mentioned in equation 3.

### A.2.2 SPATIAL FACTORS

The low-dimensional latent time series are mapped to the high-dimensional grid by the spatial factors $\mathbf{F} \in \mathbb{R}^{L \times D}$. The $d^{\text{th}}$ column of $\mathbf{F}$ represents the influence of the $d^{\text{th}}$ latent variable on each grid location. To effectively capture the correlation between spatially proximate grid points under a single latent variable, we model the spatial factors using radial basis functions (RBFs), following Manning et al. (2014); Farnoosh & Ostadabbas (2021). RBFs not only ensure locality, but they are also smooth functions that are parameter-efficient. We assume that the center parameter $\boldsymbol{\rho}_d$ of each kernel is sampled from a standard normal distribution and then passed through a sigmoid function to obtain normalized outputs between $[0, 1]$. The scale parameter $\boldsymbol{\gamma}_d$ comes from a standard normal distribution. Mathematically,

$$\boldsymbol{\rho}_d = \sigma(\mathcal{N}(0, I)), \quad \boldsymbol{\gamma}_d \sim \mathcal{N}(0, I), \quad (19)$$

$$\mathbf{F}_d^\ell = \mathrm{RBF}_d(x_\ell; \boldsymbol{\rho}_d, \boldsymbol{\gamma}d) = \exp \left( -\frac{\|x_\ell - \mathrm{sigmoid}(\boldsymbol{\rho}_d)\|^2}{\exp(\boldsymbol{\gamma}_d)} \right), \quad (20)$$

where $x_\ell$ refers to the spatial coordinates of the $\ell^{\text{th}}$ grid point, and $\sigma(\cdot)$ denotes the sigmoid function.

To capture more complex spatial structures, we model the scale $\boldsymbol{\gamma}_d$ by introducing two additional parameter matrices $\mathbf{A}$ and $\mathbf{B}$. The matrix $\mathbf{A} = \begin{bmatrix} a & b \\ c & d \end{bmatrix}$ and the vector $\mathbf{B} = \begin{bmatrix} e \\ g \end{bmatrix}$ together influence the covariance structure of the RBF. Specifically, the covariance matrix $\Sigma$ is constructed as:

$$\Sigma = \mathbf{A}\mathbf{A}^T \circ \exp(\mathbf{B}), \quad (21)$$

where $\circ$ denotes the element-wise (Hadamard) product, and $\exp(\mathbf{B}) = \begin{bmatrix} \exp(e) & 0 \\ 0 & \exp(g) \end{bmatrix}$ ensures a positive-definite structure of $\Sigma$.

This covariance structure enables the RBF to capture anisotropic scaling in different directions. The matrix $\mathbf{A}\mathbf{A}^T$ provides a base covariance matrix, while the exponential transformation of $\mathbf{B}$ ensures that the resulting matrix is positive definite. As a result, the RBF kernel, which determines the spatial factor $\mathbf{F}$, is defined as:

$$\mathbf{F}_d^\ell = \exp \left( -\frac{1}{2} \|x_\ell - \boldsymbol{\rho}_d\|_{\Sigma^{-1}}^2 \right), \quad (22)$$

where $\|x_\ell - \boldsymbol{\rho}_d\|^2_{\Sigma^{-1}} = (x_\ell - \boldsymbol{\rho}_d)^T\Sigma^{-1}(x_\ell - \boldsymbol{\rho}_d)$ represents a Mahalanobis distance, allowing the RBF to have a more sophisticated shape that depends on the learned covariance $\Sigma$.

### A.2.3 TRAINING DETAILS

We train the SPACY model for 700 epochs, using an 80/20 training and validation split to evaluate the validation likelihood during training. To prelocal minima caused by performing causal discovery on poorly inferred latent representations.

**Freezing Latent Causal Modules.** To stabilize the training and ensure accurate causal discovery, we freeze the parameters of the latent SCM and causal graph, and only train the spatial factors and encoder for the first 200 epochs. This allows the spatial factor parameters to be learned without interference from incorrect causal relationships in the latent space. Once these modules are unfrozen after 200 epochs, the complete forward model and variational distribution parameters are trained jointly for the remaining 500 epochs.

This approach ensures that the inferred latent representations are sufficiently robust before learning the causal structure of the latent SCM.

### A.2.4 EVALUATION DETAILS

The mean correlation coefficient (MCC) is adapted as a measure of alignment between the inferred and true latent variables, widely used in causal representation learning works (Yao et al., 2022b;a). Here, MCC is computed as the mean of the correlation coefficients between each pair of true and inferred latent variables, providing a balanced metric that captures how well the inferred variables match the true underlying causal structure.

To evaluate the accuracy of inferred causal graphs and representations, we match the nodes of the inferred graph to the ground truth using a permutation-invariant approach. Specifically, we apply the Hungarian algorithm to find the optimal permutation of nodes that aligns the inferred graph's adjacency matrix with the ground truth, minimizing the discrepancies between them. This optimal permutation is then used to calculate both the F1 Score and the Mean Correlation Coefficient (MCC), providing consistent node alignment across these metrics.

### A.3 SYNTHETIC EXPERIMENT

This section provides more details about how we set up and run experiments using SPACY on synthetic datasets.

### A.3.1 DATASET GENERATION

The spatial decoder, represented by the function $g_\ell$, is configured either as linear or nonlinear, depending on the experiment setting. For nonlinear scenarios, we use randomly initialized MLPs. We generate $N = 100$ samples of data, with $T = 100$ time length each and represented on a grid of size $100 \times 100$. This brings the total data dimension of $100 \times 100 \times 100 \times 100$. We vary the number of nodes ($D = 10, 20$ and $30$) in each setting.

For ground-truth latent, we generate two separate sets of synthetic datasets: a linear dataset with independent Gaussian noise and a nonlinear dataset with history-dependent noise modeled using conditional splines Durkan et al. (2019). We generate one random graph (specifically, Erdős-Rényi graphs) and treat them as ground-truth causal graphs.

**Latent: Linear SCM** We model the data as:

$$\mathbf{Z}_d^{(t)} = \sum_{k=0}^{\tau} \sum_{d'=1}^{D} (\mathbf{G} \circ W)_{d',d}^k \times \mathbf{Z}_{t-k}^d + \eta_d^t \tag{23}$$

with $\eta_d^t \in \mathcal{N}(0, 0.5)$. Each entry of the matrix $W$ is drawn from $U[0.1, 0.5] \cup U[-0.5, -0.1]$

**Latent: Non-linear SCM**   We model the data as:

$$\mathbf{Z}_d^{(t)} = f_d \left( \mathrm{Pa}_G^d(< t), \mathrm{Pa}_G^d(t) \right) + \eta_d^{(t)}$$

where $f_d$ are randomly initialized multi-layer perceptions (MLPs), and the random noise $\eta_d^{(t)}$ is generated using history-conditioned quadratic spline flow functions (Durkan et al., 2019).

**Spatial Factors**   To generate the spatial factor matrices $\mathbf{F}$, we first sample the centers $\boldsymbol{\rho}_d$ of the RBF kernels uniformly over the grid while enforcing a minimum distance constraint to ensure separation between centers. Specifically, the minimum distance between any two centers is set to be $\frac{1}{10}$ of the grid dimension. The scales $\boldsymbol{\gamma}_d$ are sampled to define the extent of each RBF kernel, drawn uniformly from the range $U[3, 6]$. With these parameters, each entry of the spatial factor matrix $\mathbf{F}_d^\ell$ is determined by the RBF kernel as follows:

$$\mathbf{F}_d^\ell = \exp \left( -\frac{||x_\ell - \boldsymbol{\rho}_d||^2}{\exp(\boldsymbol{\gamma}_d)} \right),$$

where $x_\ell$ denotes the spatial coordinates of the $\ell^{\text{th}}$ grid point, $\boldsymbol{\rho}_d$ is the center, and $\boldsymbol{\gamma}_d$ is the scale of the $d^{\text{th}}$ latent variable.

**Spatial Mapping**   For the generation of $\mathbf{X}_\ell$, we pass the product of the spatial factors and the latent time series through a non-linearity $g_\ell$:

$$\mathbf{X}_\ell = g_\ell \left( [\mathbf{FZ}]_\ell \right) + \varepsilon_\ell, \quad \varepsilon_\ell \sim \mathcal{N}(0, \sigma_\ell^2 I) \tag{24}$$

where $g_\ell$ is the spatial mapping. It is implemented as a randomly initialized multi-layer perception (MLP) with the embedding of dimension 1 in the non-linear map setting, or as an identity function in the linear map setting. $\varepsilon_\ell$ is the grid-wise Gaussian noise added.

**Baselines**   For all baselines, the default hyperparameter values are used. For Mapped-PCMCI, we referred to the implementation by (Tibau, 2022)[2]. For Linear-Response we refer to the implementation by (Falasca et al., 2024)[3]For LEAP and TDRL, the convolution neural network encoder and decoder are chosen as this architecture fits our data's modality. For LEAP we followed closely with the CNN encoder and Decoder architecture for the mass-spring system experiment, implementation details can be viewed here (Yao et al., 2022b) [4]. For TDRL we followed closely with the CNN encoder and Decoder architecture for the modified cartpole environment experiment with implementation details here (Yao et al., 2022a)[5].

### A.3.2 QUALITATIVE RESULTS

Figure 10 demonstrates our model's performance with the comparison between ground truth and inferred spatial factors $F$. Overall the modes from inferred spatial factors align well with the ground truth in terms of centers and scales, with minor deviations in shape. As the latent SCM becomes non-linear, the model shows some slight errors with at most 1 missing mode, maintaining the overall spatial representation recovery. This is also reflected by the quantitative results as performance falls slightly short for non-linear SCM.

### A.3.3 VISUALIZATION DETAILS

In this section, we describe the visualization process of spatial factors for both synthetic and Global temperature experiments, which aims to represent the spatial influence of different modes on a grid by highlighting the areas where certain modes are active. The method identifies significant regions in the grid by applying a threshold based on a chosen percentile of the weights (for example, 95%). This thresholding helps to isolate areas where a mode's spatial influence is particularly strong, creating a mask that highlights these regions.

---

[2]Mapped-PCMCI: https://github.com/xtibau/savar

[3]Linear-Response: https://github.com/FabriFalasca/Linear-Response-and-Causal-Inference

[4]LEAP: https://github.com/weirayao/leap

[5]TDRL: https://github.com/weirayao/tdrl

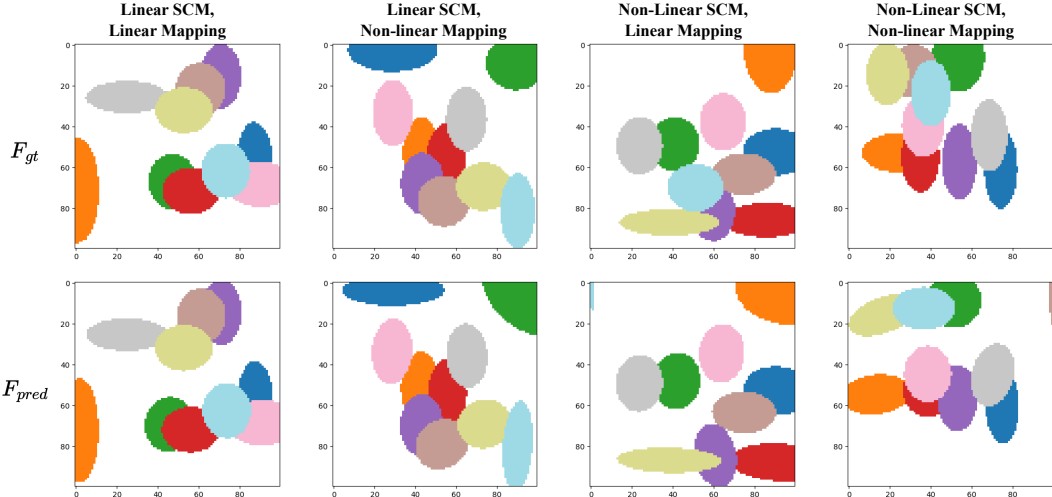

Figure 10: Visualization of the ground-truth and inferred spatial factors for different combinations of linear and nonlinear functions for SCMs and spatial mappings (top row: ground-truth, bottom row: predicted/inferred). We demonstrate the visualization when latent dimension $D = 10$
.

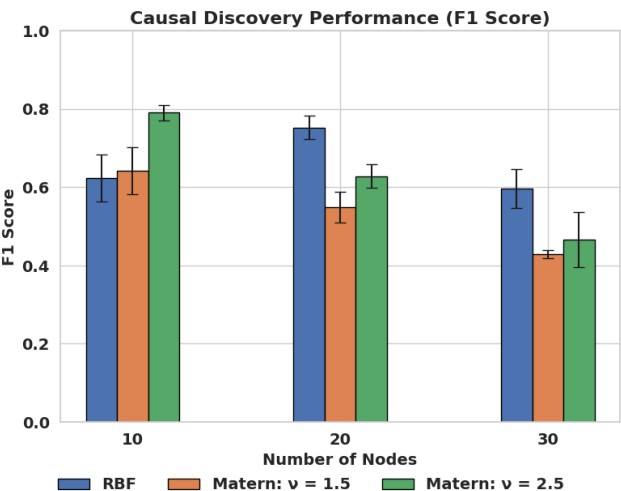

Figure 11: The causal discovery performance (F1 score) of SPACY using different kernel functions as spatial factors. Average of 5 seeds reported

These masked regions are then combined to generate a comprehensive view of how all modes influence the spatial grid. The visualization distinguishes the areas affected by different modes, allowing for easy identification of their spatial patterns and overlaps. This approach allows for a clear visual interpretation of the complex spatial structure represented by the modes, facilitating the analysis of their respective influences and interactions.

For complex spatial factors and graphs, we use a merging process that simplifies the causal global dynamics by combining nodes based on the proximity of node centers. The process identifies merging clusters in the grid by applying a threshold based on a chosen percentile of all the pair-wise distances (for example, lower $5\%$), and merging nodes that fall below the threshold.

## A.4 ABLATION STUDY

**Different Kernels** To assess the robustness and generalizability of SPACY's variational inference framework, we experiment with different kernel functions in modeling spatial-temporal dynamics. We use the synthetic dataset with linear SCM and nonlinear spatial mapping.

The Matérn kernel is a generalization of the RBF kernel that introduces an additional parameter $\nu$ controlling the smoothness of the function. By adjusting $\nu$, the Matérn kernel can model functions with varying degrees of smoothness, providing more flexibility than the RBF kernel. We test SPACY with the Matérn kernel using two settings: $\nu = 1.5$ and $\nu = 2.5$.

We replace the RBF kernel in SPACY with the Matérn kernel using $\nu = 1.5$ and $\nu = 2.5$. The inferred spatial modes' general locations and scales align well with the ground truth across all kernel settings (illustrated in Figure 12). This consistency demonstrates that SPACY's spatial representations are robust to the choice of kernel function.

Figure 11 presents the F1-Score and MCC for SPACY using the RBF kernel and both Matérn kernel settings. The results show that SPACY achieves similar or even competitive performance with the Matérn kernels compared to the RBF kernel, indicating that the variational inference framework effectively generalizes across different kernel functions.

The Matérn kernel is a generalization of the Radial Basis Function (RBF) kernel and is widely used in spatial statistics and machine learning due to its flexibility in modeling functions of varying smoothness. The Matérn kernel is defined as:

$$k_{\text{Matérn}}(r) = \frac{2^{1-\nu}}{\Gamma(\nu)} \left( \sqrt{2\nu} \frac{r}{\ell} \right)^{\nu} K_{\nu} \left( \sqrt{2\nu} \frac{r}{\ell} \right),$$

where:

- $r = \|\mathbf{x} - \mathbf{x}'\|$ is the Euclidean distance between points $\mathbf{x}$ and $\mathbf{x}'$,
- $\ell$ is the length scale,
- $\nu > 0$ controls the smoothness of the function,
- $\Gamma(\cdot)$ is the gamma function,
- $K_{\nu}(\cdot)$ is the modified Bessel function of the second kind.

For specific values of $\nu$, the Matérn kernel simplifies to closed-form expressions:

- **When $\nu = 1.5$:**
$$k_{\text{Matérn}}^{1.5}(r) = \left( 1 + \frac{\sqrt{3}r}{\ell} \right) \exp\left( -\frac{\sqrt{3}r}{\ell} \right).$$

- **When $\nu = 2.5$:**
$$k_{\text{Matérn}}^{2.5}(r) = \left( 1 + \frac{\sqrt{5}r}{\ell} + \frac{5r^2}{3\ell^2} \right) \exp\left( -\frac{\sqrt{5}r}{\ell} \right).$$

These formulations allow us to model functions with different degrees of smoothness, providing a more flexible approach compared to the RBF kernel.

From the visualization in 12 when $D = 10$, despite changing the kernel function type, the modes from inferred spatial factors align well with the ground truth in terms of location and scale. This suggests that SPACY is robust to the kernel choice in modeling the spatial factors.

A.5 HYPERPARAMETER DETAILS

In this section, we list the hyperparameters choices for SPACY in our experiments.

For our SPACY model, we used an augmented Lagrangian training procedure to enforce the acyclicity constraint in the model (Zheng et al., 2018). We closely follow the procedure employed by Gong Wenbo & Nick (2022) for scheduling the learning rates (LRs) across different modules of our model.

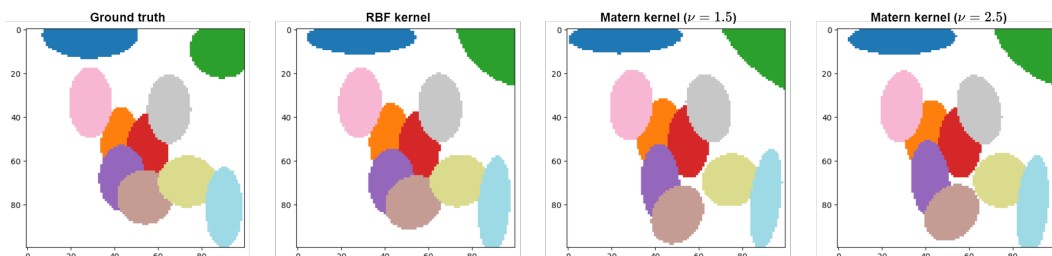

Figure 12: Overview of the visualization of the spatial factor when using different kernel functions. We compare inferred spatial factors using RBF, Matern Kernel ($\nu = 1.5$), and Matern Kernel ($\nu = 2.5$) with the ground truth spatial factors

| Dataset | Synthetic-L ($D = 10, 20, 30$) | Synthetic-NL | Global Temperature |
|---|---|---|---|
| **Hyperparameter** | | | |
| Matrix LR | $10^{-3}$ | $10^{-3}$ | $10^{-3}$ |
| SCM LR | $10^{-3}$ | $10^{-3}$ | $10^{-3}$ |
| Spatial Encoder LR | $10^{-3}$ | $10^{-3}$ | $10^{-3}$ |
| Spatial Factor LR | $10^{-2}$ | $10^{-2}$ | $10^{-2}$ |
| Spatial Decoder LR | $10^{-3}$ | $10^{-3}$ | $10^{-3}$ |
| Batch Size | 100 | 100 | 100 |
| # Outer auglag steps | 60 | 60 | 60 |
| # Max inner auglag steps | 6000 | 6000 | 6000 |
| $f_\ell$ embedding dim | none | 64 | none |
| Sparsity factor $\lambda$ | 10 | 10 | 10 |
| Spline type | None | Quadratic | None |
| $g_\ell$ embedding dim | 32 | 32 | 32 |

Table 1: Table showing the hyperparameters used with SPACY.

For the Synthetic-L, Synthetic-NL, and Global Temperature datasets, the outer augmented Lagrangian (auglag) steps are set to 60, with a maximum of 6000 inner auglag steps. This provides an effective balance between model convergence and training efficiency, ensuring thorough exploration of the parameter space without premature stopping.

We used the rational spline flow model described in Durkan et al. (2019). We use the quadratic or linear rational spline flow model in all our experiments, both with 8 bins. The MLPs $f_\ell$ and $g_\ell$ have 2 hidden layers each and LeakyReLU activation functions, where $e$ is the embedding dimension. We also use layer normalization and skip connections. Table 1 summarizes the hyperparameters used for training.

### A.6 GLOBAL TEMPERATURE

The **Global Temperature Dataset** is a comprehensive, mixed real-simulated dataset encompassing monthly global temperature data spanning the years 1999 to 2001. It contains 7531 simulated samples, each with a time sequence of 24 months, covering the entire globe at a fine spatial resolution. The grid size is $145 \times 192$, which corresponds to a spatial division of approximately $1.24°$ latitude and $1.875°$ longitude. This spatial resolution allows the dataset to provide detailed global coverage, capturing temperature variations across diverse geographical regions. The resulting data dimensions are $7531 \times 24 \times 145 \times 192$, representing the total number of samples, the temporal sequence, and the spatial grid, respectively.

To facilitate causal analysis of complex climate phenomena beyond seasonal patterns, a deseasonality procedure was applied. This normalization process involved computing the monthly mean for each month across all years and then adjusting the data accordingly (for example, normalizing all January data by the mean of all January values). This approach aims to remove regular

seasonal influences, thereby emphasizing more intricate climate events and enabling deeper causal learning and understanding of global temperature dynamics.

For our analysis, we employ the SPACY method to uncover latent representations within the data. These representations capture regions of similar weather properties and help identify causal links between these regions and weather phenomena occurring elsewhere. The methodology uses a linear functional relationship paired with multi-layer perceptron (MLP) spatial decoding. Specifically, we use 25 latent variables (denoted as $D = 25$) and a maximum lag of three months ($\tau = 3$).

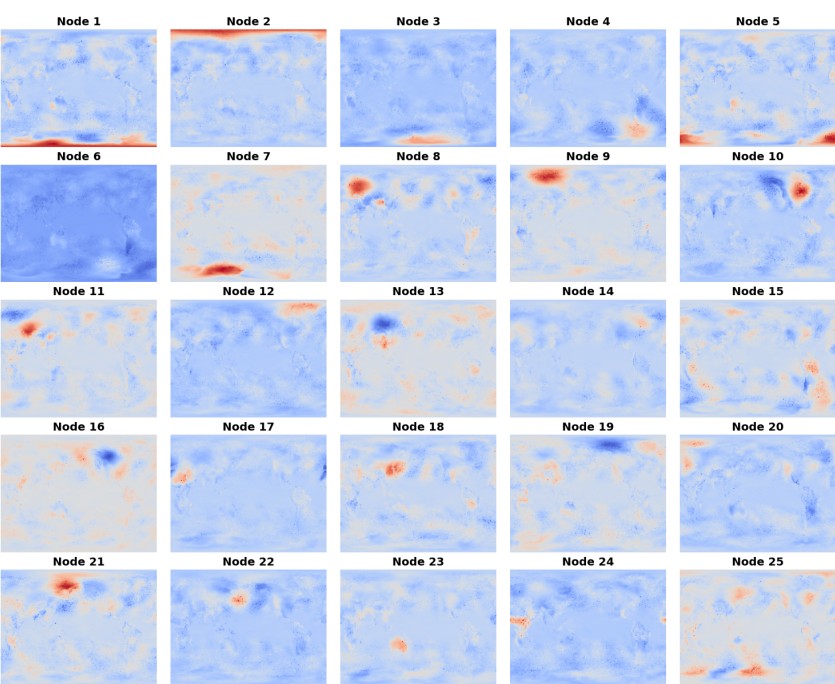

Figure 13: Visualization of the spatial nodes inferred by Varimax-PCA from the Global Temperature Dataset

Figure 13 demonstrates the visualization of the individual modes being reduced from Varimax-PCA. The method did a decent interpolation as some nodes/components exhibit clear spatial patterns that are interpretable in terms of physical or location-based information. However, multiple components are more diffuse and have less interpretable locations. For instance, it may be hard to attribute physical location for node $13, 14, 19, 25$. There are also clusters of nodes that show similar spatial features, such as node $4, 6$, suggesting they capture similar underlying components.

The visualization of the modes and causal graph deduced by Mapped-PCMCI is shown in Figure 14. While the locality pattern can be observed in important regions such as Australia, Africa, and East Asia, many of the inferred modes appear diffused across the map. This suggests that the underlying spatial structure is not cleanly partitioned into distinct, interpretable modes.

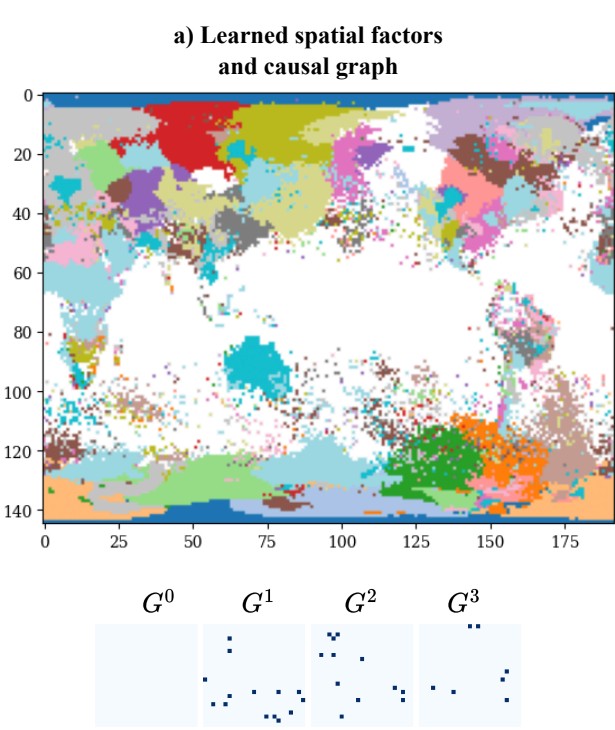

Figure 14: Visualization of the spatial factor inferred by Varmax-PCA and causal graph inferred by PCMCI+, following the procedure in section A.3.3

