# OpenReview forum: "Discovering Latent Structural Causal Models from Spatio-Temporal Data"
_ICLR.cc/2025/Conference — ICLR 2025 Conference Withdrawn Submission_

### Official Review · Reviewer_U4fy · 2024-10-25

**Soundness:** 1
**Presentation:** 1
**Contribution:** 2
**Rating:** 3
**Confidence:** 5

**Summary:**

The work proposes a variational inference based framework for estimating the latent variables and underlying causal relationships in spatiotemporal data. The authors provide theoretical claims about their approach, and empirically evaluate the method against other causal discovery baselines on synthetic and real world climate data.

**Strengths:**

The paper considers an interesting problem.

**Weaknesses:**

The most fundamental weakness lies in the theory. The proofs of Theorem 1 and 2 are incorrect. Specifically:
1. Equal in distribution does not mean two random variables are equal. The first step of the proof for theorem 1 is problematic.
2. Similarly, the proof of theorem 2 is incorrect because we cannot deduce Z = M\tilde{Z} from their probability distributions.
3. I don’t think the current proof can be made correct. In general, the linear identifiability requires supervision [1], and the permutation identifiability requires additional constraints on the generative process [2]. It is unclear how the setting considered here is relevant with these.
4. Without permutation identifiability, the inferred latent factors are actually arbitrarily entangled. The linear identifiability is insufficient for the use case.

At this point, I don’t think additional discussions are needed.

[1] Khemakhem, Ilyes, et al. "Variational autoencoders and nonlinear ica: A unifying framework." International conference on artificial intelligence and statistics. PMLR, 2020.

[2] Lachapelle, Sébastien, et al. "Disentanglement via mechanism sparsity regularization: A new principle for nonlinear ICA." Conference on Causal Learning and Reasoning. PMLR, 2022.

**Questions:**

My overall suggestion is that, please refer to [1] for correct definitions of identifiability for variational inference models, and correct ways to prove identifiability.

---

### Official Review · Reviewer_oPzv · 2024-10-28

**Soundness:** 3
**Presentation:** 3
**Contribution:** 2
**Rating:** 6
**Confidence:** 3

**Summary:**

This study focuses on causal discovery in spatiotemporal data, which is an important scientific task in several scientific domains. The authors propose a variational inference based framework to explicitly model latent time-series and their causal relationships from spatially confined modes in the data. experimentally and theoretically validated for its effectiveness.

**Strengths:**

1.The paper focuses on the issue of causal discovery in spatial-temporal data , which is a fascinating and significant area of research.

2.The authors have conducted experiments on both synthetic and real-world datasets, validating the effectiveness of the model.

3. The authors have provided the source code, which ensures good reproducibility.

**Weaknesses:**

1.The authors propose the “Spatial Factor” for modeling spatial dependencies, while it is questionable whether a single matrix can effectively model complex spatial relationships. The authors need to provide further clarification or stronger evidence to support their claims.

2.The authors lack discussion of some existing work, for example, the paper [1] also focus on causal discovery in spatial-temporal data. It is essential for authors to further discuss the similarities and differences between the two papers and alleviate my concerns about the novelty of this work.

[1] Zhao, Yu, et al. "Generative Causal Interpretation Model for Spatio-Temporal Representation Learning." Proceedings of the 29th ACM SIGKDD Conference on Knowledge Discovery and Data Mining. 2023.

**Questions:**

Please see the weaknesses.

---

### Official Review · Reviewer_cgLU · 2024-11-04

**Soundness:** 3
**Presentation:** 2
**Contribution:** 2
**Rating:** 5
**Confidence:** 3

**Summary:**

In this paper, the authors propose SPACY which learns a causal graph for spatio-temporal data. This method aims to solve two problems associated with the previous method: 1) the high dimensionality of the spatio-temporal data; 2) correlations between spatially proximate points may bury the true causal relationships. To this end, we propose to first cluster the spatially adjacent points (denoted as a common spatial factor) and then infer the causal graph between the spatial factors. They prove the identifiability and use empirical experiments to validate their method.

**Strengths:**

1. The problem is well formulated and the method is applicable to relatively high-dimensional data
2. The method can infer both lagged and instantaneous causal links
3. The proof of the identifiability is essential

**Weaknesses:**

1. The authors don't seem to mention how to guarantee that the learned graph G is a DAG, especially for the instantaneous relations.
2. The experiments are conducted on data with regular grids. How about irregular grids?
3. It may be more useful if the authors can further prove that the causal graph itself is identifiable.

**Questions:**

1. In the synthetic experiments, the generative model is assumed to follow the model assumptions and this explains the good performance of SPACY. However, for real-world data, the dependencies can be very complicated. How do we argue that such assumptions still hold? For example, there may still exist conditional dependencies between spatially proximate points, even conditioned on the common spatial factors and the causal graphs. Moreover, the instantaneous graph may not be causal, but undirected and cyclic instead. If the assumptions do not hold, can SPACY still offer some insight?
2. What about the time-varying case? I think the time-varyinig assumption is more realistic. Perhaps the authors can include a discussion on how to extend SPACY to time-varying scenarios.
3. Using variational inference to deal with such complicated latent space can be slow and may get stuck in local optima. It's better to discuss how the training time scales with the size of the latent space and how to find a good initial point for optimization / how to escape from the local optima.

---

### Official Review · Reviewer_3RFp · 2024-11-05

**Soundness:** 2
**Presentation:** 2
**Contribution:** 3
**Rating:** 3
**Confidence:** 4

**Summary:**

This paper introduces a framework for causal discovery in latent variable models on spatiotemporal data. It presents a very flexible model where the latent variables are driven by SCMs defined on non-stationary time series, which are mapped to high-dimensional observations via an RBF kernel which addresses spatial correlations. The approach outperforms current approaches in spatiotemporal settings, and it is well-motivated on real climate data. However, the identifiability of the generative process is very weak and not acknowledged properly in the main text.

**Strengths:**

- The paper proposes a novel framework for learning causally-driven spatio-temporal data.
- The estimation method outperforms other current approaches for learning sequential latent variable models in both MCC and causal graph estimation.
- It demonstrates a very interesting application and scalability to realistic scenarios with earch climate systems.

**Weaknesses:**

The main weakness of the paper relies on **identifiability**. Below are all my concerns with regards to this matter. Other concerns will be written in the **Questions** part as I believe they are less important.
- Identifiability is introduced without a proper definition in the introduction (it appears in line 67 for the first time).
  - Identifiability should be defined along with corresponding citations to understand what is the equivalence you are looking for in your problem (I would assume graph structure).
  - It should also be accompanied with motivation on why identifiability is important, e.g. proving you have a unique solution up to some equivalence, scientific discovery (assuming assumptions in real setups hold), etc.
- Identifiability results are very weak considering the complexity of the data generating process. It only considers linear mixing which gives some affine identifiability (correct if mistaken), which feels redundant.
  - Line 315: "We focus on the specific case where no non-linearity maps the latents to the observable space". Then, Theorem 2 establishes identifiability up to " transformation by an invertible matrix". The result does not feel very strong as it does not seem to go beyond linear mixing.
  - Also given $Z\in\mathbb{R}^{D\times T}$, which is identifiable up to an invertible transformation, does this mean that theres mixing no only on $D$, but also over $T$? This should be clarified in the main text.
- Identifiability does not cover all the target latent variables:
  - Line 174: “Our goal is to infer the latent variables and the true causal graph in an unsupervised manner”. However, no identifiability results with respect to graph structure.
  - Generally I don't consider the graph structure to be identifiable considering identifiability up to a linear transformation (correct me if I am mistaken). Even with permutation and scaling identifiability, I don't see how instantaneous effects can be discovered with your assumptions.
  - For latent graph estimation:
    - In the absence of instantaneous effects, I would expect first identifiability of the latent variables up to permutation and component-wise nonlinearity.
    - With instantaneous effects, the task is even more challenging. Zhang et al 2024 recently has some results in non-temporal data.
- The paper generally does not acknowledge the above limitations, which I find an important concern given the attention it gives to the theoretical part.
  - Line 69: “our framework can handle both instantaneous edges and overlapping spatial factors”. Although it might be true empirically, this is not true with regards to identifiability. Therefore, I don't consider this claim to be valid.
  - Line 532: "We discussed the structural identifiability of our model". Structural identifiability as defined by Peters et al. 2017, refers to identifiability of the causal graph. Therefore, this is not true if the above points are correct.

**Note:** I would be happy to raise my score provided that this issue is addressed either by acknowledging this limitation in comparison to other works or establishing some stronger identifiability results (at least for the causal graph).

**References**

Zhang, K., Xie, S., Ng, I. &amp; Zheng, Y.. (2024). Causal Representation Learning from Multiple Distributions: A General Setting. Proceedings of the 41st International Conference on Machine Learning, in Proceedings of Machine Learning Research 235:60057-60075 Available from https://proceedings.mlr.press/v235/zhang24br.html.

Peters, J., Janzing, D., & Schölkopf, B. (2017). Elements of causal inference: foundations and learning algorithms (p. 288). The MIT Press.

**Questions:**

- The **preliminary** paragraph appears in the related work section, but it feels more like background.
- Problem setting: “ These L time series are arranged in a K-dimensional grid G” is confusing, as in the notation K does not appear and it is clear how it involves X and Z. It would be better to write an example for your case: $L=\sqrt{L} \times \sqrt{L}$. (please correct this if it was misunderstood).
- Notation: Given that $F$ is a matrix, wouldnt it be better to write $F_{ld}$ instead of $F^l_d$ for readability?
- In experiments, how is it possible to achieve high MCC scores even with identifiability up to an invertible matrix? From your explanations, it does not seem that you perform any alignment with respect to a linear transformation.

---

### Author Response · Authors · 2024-11-28

# Response to the Reviewers

We sincerely appreciate the reviewers' thoughtful feedback and detailed critiques, which have offered valuable insights for improving our work. We acknowledge the inherent challenges of establishing rigorous theoretical guarantees in the complex domain of spatiotemporal causal discovery. We are grateful for the reviewers' helpful suggestions regarding identifiability analysis and will strive to address the concerns raised, working to establish stronger results in this area.

---

### Note · Authors · 2024-11-28

**Comment:**

# Response to the Reviewers

We sincerely appreciate the reviewers' thoughtful feedback and detailed critiques, which have offered valuable insights for improving our work. We acknowledge the inherent challenges of establishing rigorous theoretical guarantees in the complex domain of spatiotemporal causal discovery. We are grateful for the reviewers' helpful suggestions regarding identifiability analysis and will strive to address the concerns raised, working to establish stronger results in this area

**Withdrawal Confirmation:**

I have read and agree with the venue's withdrawal policy on behalf of myself and my co-authors.